# Cellular diversity in the *Drosophila* midbrain revealed by single-cell transcriptomics

**Vincent Croset[†], Christoph D Treiber[†]\*, Scott Waddell\***

Centre for Neural Circuits and Behaviour, The University of Oxford, Oxford, United Kingdom

**Abstract** To understand the brain, molecular details need to be overlaid onto neural wiring diagrams so that synaptic mode, neuromodulation and critical signaling operations can be considered. Single-cell transcriptomics provide a unique opportunity to collect this information. Here we present an initial analysis of thousands of individual cells from *Drosophila* midbrain, that were acquired using Drop-Seq. A number of approaches permitted the assignment of transcriptional profiles to several major brain regions and cell-types. Expression of biosynthetic enzymes and reuptake mechanisms allows all the neurons to be typed according to the neurotransmitter or neuromodulator that they produce and presumably release. Some neuropeptides are preferentially co-expressed in neurons using a particular fast-acting transmitter, or monoamine. Neuromodulatory and neurotransmitter receptor subunit expression illustrates the potential of these molecules in generating complexity in neural circuit function. This cell atlas dataset provides an important resource to link molecular operations to brain regions and complex neural processes.

DOI: https://doi.org/10.7554/eLife.34550.001

**\*For correspondence:**
christoph.d.treiber@gmail.com (CDT);
scott.waddell@cncb.ox.ac.uk (SW)

[†]These authors contributed equally to this work

**Competing interests:** The authors declare that no competing interests exist.

## Introduction

Neuroscience is typically studied at the systems, cellular, or molecular level. However, it will be necessary to bridge these traditional boundaries to fully understand how the brain operates. Such a momentous task is somewhat simplified if analyses are focused on an animal with a relatively small brain, but where systems-level processes are evident. In many respects, the vinegar fly *Drosophila melanogaster* fits the bill (*Haberkern and Jayaraman, 2016*). *Drosophila* have an estimated 150,000 neurons in the entire brain, of which the optic lobes, or visual neuropils, comprise two thirds of this neural mass. The remaining approximately 50,000 neurons, or midbrain, houses many key neural structures such as the mushroom bodies and central complex, which are, amongst other things, critical for memory-directed behavior (*Cognigni et al., 2018*) and navigation (*Seelig and Jayaraman, 2015*), respectively.

Recent large-scale electron-microscopy projects have generated wiring diagrams, or connectomes, of parts of the larval and adult fly nervous system (*Berck et al., 2016*; *Eichler et al., 2017*; *Ohyama et al., 2015*; *Takemura et al., 2013*; *Takemura et al., 2017a*; *Takemura et al., 2017b*; *Tobin et al., 2017*; *Zheng et al., 2017*). While these efforts are an essential part of the quest to decipher brain function, they are not enough. Genes determine the anatomy and mode of connectivity, the biophysical properties, and the information-processing limits of individual constituent neurons. Therefore, understanding any given wiring diagram requires a systematic view of gene expression within their functionally relevant cellular context. With this knowledge in hand, investigators can begin to examine how gene products contribute to cell- and circuit-specific functions and, ultimately, organismal behavior.

New developments in single-cell sequencing technology provide a unique means to generate such a brain-wide view of gene expression with cellular resolution. Massively parallel approaches, such as Drop-seq (*Macosko et al., 2015*), permit simultaneous analysis of the transcriptomes of 1000 s of individual cells. In brief, each cell from a dissociated tissue is first captured with an oligonucleotide bar-coded bead in a nanoliter aqueous droplet. Inside each droplet, the same cell identifier sequence becomes attached to all mRNA molecules from an individual cell. Following this critical cell-specific hybridization step, all the material from 1000 s of individual cells can be pooled and processed together for mRNA sequencing. Drop-seq therefore provides the means to access the transcriptomes of a representation of most cells in the fly midbrain.

A key hurdle in generating a single-cell atlas of the brain is the ability to assign individual transcriptome profiles to the correct cell, or at least cell-type. Again, using an animal whose brain has an intermediate number of neurons and presumably neural diversity simplifies the task. Moreover, years of genetic analyses in *Drosophila* have provided a considerable number of established transgenic and intrinsic markers for specific brain regions and cell-types. These identifiers often allow one to extract the relevant cell profiles from the larger dataset.

Here we report the application and an initial analysis of Drop-seq data to investigate the cellular diversity of the *Drosophila* midbrain. We demonstrate the ability to assign many single-cell profiles to identified cell-types and brain regions, and identify novel markers for these regions. Moreover, cells can be robustly classified based on their neurotransmitter profile. We find that certain neuropeptides preferentially accompany particular fast-acting transmitters, or monoamines. In addition, we detail the apparent complexity of modulatory and neurotransmitter receptor subunit expression. This single-cell dataset provides an indication of the extent of neural diversity in the fly brain, and provides essential cellular context linking molecules to neural circuits and brain function.

## Results

### Drop-seq analysis of the *Drosophila* midbrain

We first optimized the conditions required to effectively dissociate and capture individual *Drosophila melanogaster* cells with DNA bar-coded microparticles in aqueous droplets, using a commercially available apparatus. *Drosophila* neurons are about a tenth of the size of mammalian cells. We therefore first verified the efficiency of processing insect cells and of single-cell capture by generating single-cell transcriptomes attached to microparticles (STAMPs) from a cell suspension comprised of a 1:1 mixture of *Drosophila* S2 and *Spodoptera frugiperda* (fall armyworm) Sf9 cultured cells. We then sequenced these S2/Sf9 STAMPs (*Figure 1—figure supplement 1A*). This procedure retrieved 764 barcode-associated transcriptomes, of which 368 were identifiable as *Drosophila* and 384 as *Spodoptera*. Importantly, only 12 transcriptomes contained cDNA coming from both species (*Figure 1—figure supplement 1B*), indicating that only 3.2% of all sequenced transcriptomes resulted from capturing two cells together. This analysis suggested that the Drop-seq system and our chosen parameters are suitable for barcoding single insect cells and are optimized to minimize capture of cell doublets.

We next used these same parameters and concentrations to collect STAMPs from thousands of cells from the *Drosophila* midbrain, in eight independent biological replicates, over eight different days. Each day we isolated single-cells from 80 to 100 dissected brains taken from an equal number of male and female flies. Brains were removed from the head capsule, optic lobes were manually dissected away and a single-cell suspension was prepared from the remaining fly midbrains. Larger tissue fragments were removed by filtration, and the eluant was inspected under a microscope to confirm the presence of single cells and the absence of clumps. Cells were individually paired with DNA barcoded beads and cDNA libraries were generated from bead-bound single-cell transcriptomes, and sequenced (*Figure 1A*) (*Macosko et al., 2015*). Pooling the data from the eight independent experiments resulted in a dataset of 19,260 cells, with each containing between 200 and 10,000 unique molecular identifiers (UMIs) and therefore, single mRNA molecules. We performed a Principal Component Analysis (PCA) on these transcriptomes and reduced the top 50 PCs into two dimensions using t-SNE (*Van Der Maaten, 2014*) (*Figure 1—figure supplement 2A*). We selected the cut-off for the optimal number of UMIs per cell to be included in our analyses by generating t-SNE plots from data with a variety of quality thresholds. These analyses revealed that discarding

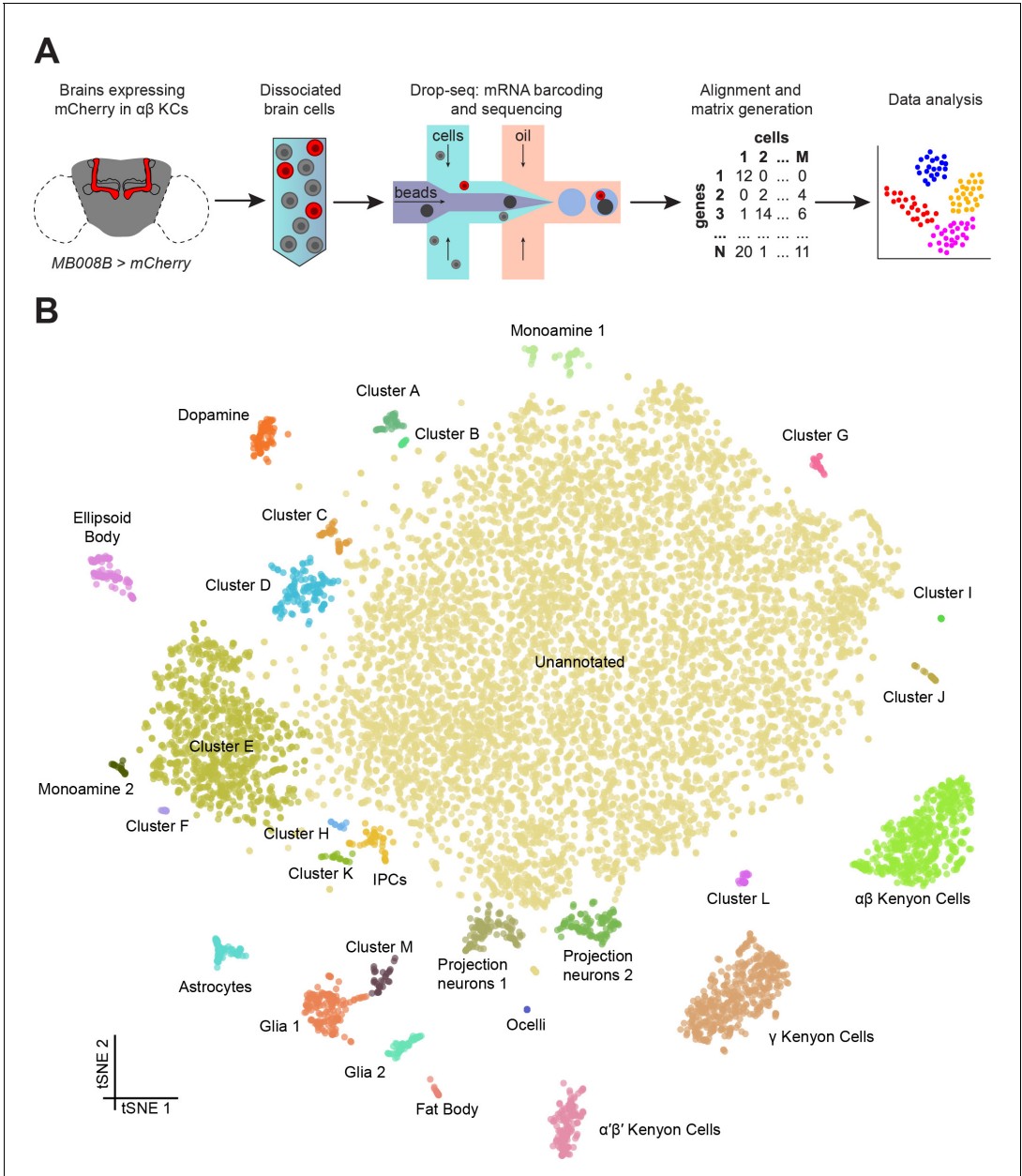

**Figure 1.** Drop-seq reveals neuronal clusters in the *Drosophila* brain. (**A**) Schematic of the experimental procedure. *Drosophila* brains were dissected and dissociated prior to Drop-seq. After sequencing and alignment, a digital expression matrix containing information about the number of UMIs found for each gene, in each cell, was generated and used for PCA and subsequent analyses. See Materials and methods section for details. (**B**) Two-dimensional representation (t-SNE) of 10,286 *Drosophila* brain cells, manually classified into 28 clusters. Based on the recovery of cell-types of known abundance in the brain, we estimate that there are 45,000 cells in the fly midbrain.

DOI: https://doi.org/10.7554/eLife.34550.002

The following source data and figure supplements are available for figure 1:

**Source data 1.** Digital Expression Matrix.
DOI: https://doi.org/10.7554/eLife.34550.007

**Source data 2.** List of marker genes for each cluster in *Figure 1*.
DOI: https://doi.org/10.7554/eLife.34550.008

**Figure supplement 1.** Preliminary validation of Drop-seq on insect cells.
DOI: https://doi.org/10.7554/eLife.34550.003

**Figure supplement 2.** – Comparison of different quality filters.
DOI: https://doi.org/10.7554/eLife.34550.004

*Figure 1 continued*

**Figure supplement 3.** – t-SNE plot showing all eight replicates.
DOI: https://doi.org/10.7554/eLife.34550.005
**Figure supplement 4.** – Sex-determination of individual neurons.
DOI: https://doi.org/10.7554/eLife.34550.006

cells with less than 800 UMIs, resulted in a data set of 10,286 high quality cells, segregated with k-means clustering into 29 cell clusters, with several corresponding to most of the known iterative, or large populations of, cell types in the *Drosophila* brain (*Figure 1—figure supplement 2B*). More stringent criteria decreased the number of cells included without further improving the clustering (*Figure 1—figure supplement 2C*). A comparison between our eight individual replicate experiments revealed that all of them contributed equally to all but one cluster (*Figure 1—figure supplement 3*). We therefore chose to use the 10,286 cells that have ≥800 UMIs from our eight pooled replicates for our subsequent analyses.

We assessed the transcript drop-out rate by determining the number of neurons that express the male-specific long non-coding *RNA on the X 1* (*roX1*) gene (*Kelley and Kuroda, 2003*; *Amrein and Axel, 1997*). For this analysis we excluded non-neuronal tissue, and Cluster M – see descriptions below. The distribution of neurons containing UMIs for *rox1* was biphasic, with one peak at 0, and another at 9 UMIs (*Figure 1—figure supplement 4A*). Since our data were prepared from an equal number of male and female brains we reasoned that these two cell populations correspond to neurons from female and male flies, respectively. We used the median between the two peaks (4.5) as a cut-off to separate the two populations, which revealed that 55.9% of neurons are positive for *roX1* (*Figure 1—figure supplement 4B*). Since this number is greater than 50%, this distribution suggests that drop-out of the *roX1* transcript is low in our high-quality dataset. However, given that the drop-out rate for each gene is influenced by the expression level and other factors that influence the ability to capture transcripts from the cell bodies using the polyadenylated tail, it is not possible to determine a global drop-out rate. Nevertheless, the rate of *roX1* drop-out provides a useful measure to compare data between different samples and preparation techniques. The neuronal Cluster J is almost exclusively comprised of *roX1*-negative cells. This could either mean that cells in this cluster are only present in the female brain, or that they represent a subset of *roX1* negative neurons that are present in both male and female brains.

We manually annotated 30 clusters in the t-SNE plot of 10,286 cells, with each containing between 9 and 7167 cells (*Figure 1B*). For each cluster, we identified a series of genes whose expression was significantly higher than in the rest of the brain (*Figure 1—source data 2*). We then used the published expression patterns for many of these genes to assign identity to several clusters (*Figure 1—source data 2*). This approach allowed us to identify the mushroom body Kenyon Cells (KCs), olfactory projection neurons (PNs), ellipsoid body ring neurons, monoaminergic neurons, astrocytes and other glia, and insulin producing cells (IPCs). We also identified a few cells from the ocelli, in addition to fat body tissue, some of which is present in the head capsule and therefore is also expected to be included in our dissected brain tissue. We also identified 13 additional cell clusters that we could not at this time assign to a particular neuronal type, and that we name with the letters A-M. Surprisingly, cluster G only contained cells obtained from a single replicate experiment (*Figure 1—figure supplement 3*). The largest cluster of all contains 7167 cells with a variety of expression profiles, that at this stage of analysis, we marked as 'unannotated', but that can nevertheless be segregated for example, based on their primary fast-acting neurotransmitter (see below, and Figure 5).

## Identification of mushroom body Kenyon Cells

The easiest and most certain way to assign a Drop-seq cluster to a specific cell-type is to track the expression of a transgenically expressed marker. For this reason, our single-cell expression dataset was generated from a genotype of flies that express an mCherry transgene specifically in the αβ subset of mushroom body KCs (*Figure 2A*). To our surprise, visualizing mCherry expression levels in our dataset revealed labeling of a very distinct group of cells (*Figure 2B*), that allowed us to assign this cluster to αβ KCs.

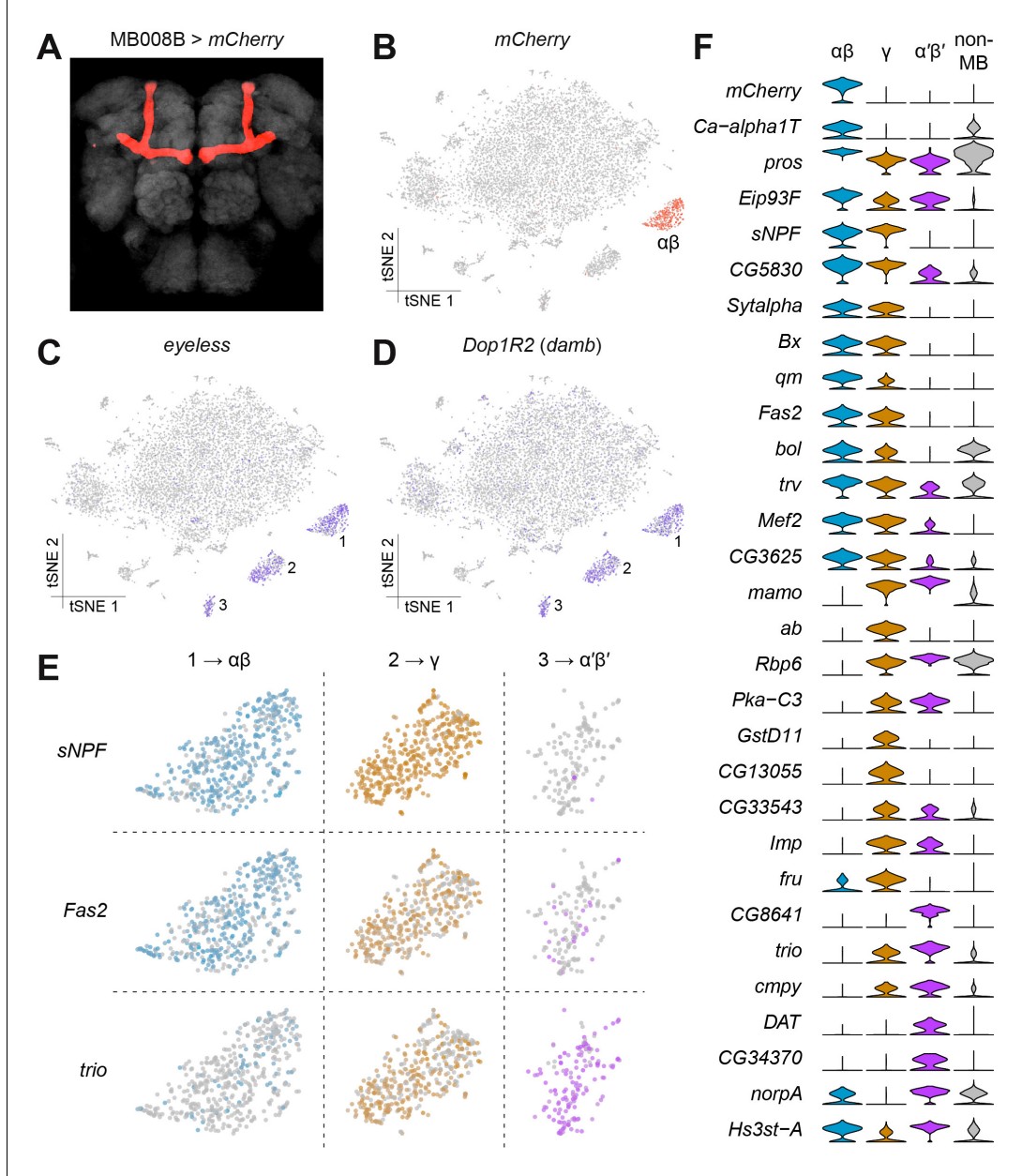

**Figure 2.** Identification of Kenyon Cells and mushroom body-specific genes. (**A**) mCherry labeling of MB008B neurons in the *Drosophila* brain. Neuropil, labeled by nc82 anti-Brp antibody, is shown in grey. (**B**) Expression of mCherry in the t-SNE-clustered brain cells shown in *Figure 1B*. mCherry-positive cells are labeled red and identify this cluster as αβ Kenyon Cells (KCs). Intensity of red (or other colors in the panels below) is proportional to the normalized expression level. (**C**) and (**D**) Expression of *eyeless* and *Dop1R2* (*damb*), in t-SNE-clustered brain cells. The three numbered clusters containing indigo cells are KCs. (**E**) Expression of *sNPF*, *Fas2* and *trio* in the three t-SNE clusters numbered in (**C**) and (**D**). Cells in light blue, orange and purple express each of these genes in αβ, γ, and α′β′ KCs, respectively. *sNPF* and *Fas2* are mostly expressed in αβ and γ KCs, while *trio* is mostly detected in γ and α′β′ KCs. (**F**) Violin plots showing the main markers that distinguish KC subtypes from each other (pairwise comparisons for genes expressed in >50% of cells in either cluster; Log2 FC >1.5, Wilcoxon rank-sum test with Bonferroni-corrected p-value<0.01). The column on the right (grey) indicates the level of expression of these genes across all non-MB neurons in the brain.

DOI: https://doi.org/10.7554/eLife.34550.009

The following source data and figure supplement are available for figure 2:

**Source data 1.** Primer pairs used for qPCR.

DOI: https://doi.org/10.7554/eLife.34550.011

**Figure supplement 1.** – Confirmation of Kenyon Cell type specific gene expression using qPCR on FAC-sorted cells.

DOI: https://doi.org/10.7554/eLife.34550.010

The mushroom bodies are brain structures that are critical for olfactory learning and memory (*de Belle and Heisenberg, 1994*; *Cognigni et al., 2018*; *Heisenberg, 2003*) and they are comprised of three main classes of neurons, the αβ, α′β′ and γ neurons, that are morphologically unique and have dissociable roles in memory processing and expression (*Crittenden et al., 1998*; *Krashes et al., 2007*; *Trannoy et al., 2011*). We first identified the α′β′ and γ KC types, using the expression of the previously known general KC markers *eyeless* and *Dop1R2* (also known as *Dopamine receptor in mushroom bodies, damb*) (*Han et al., 1996*; *Kurusu et al., 2000*). Cells expressing these two markers were contained within three distinct clusters, including the αβ cluster identified as expressing mCherry (*Figure 2C–D*). The αβ and γ KCs have previously been shown to be distinguishable from the α′β′ neurons using the expression of molecular markers. The αβ and γ KCs express *short neuropeptide F precursor* (*sNPF*) (*Johard et al., 2008*) and *Fasciclin 2* (*Fas2*) (*Cheng et al., 2001*; *Crittenden et al., 1998*), whilst α′β′ and γ KCs express the rho guanyl-nucleotide exchange factor gene *trio* (*Awasaki et al., 2000*). The expression patterns of these three genes permitted us to assign each KC cluster to one of these KC subtypes (*Figure 2E*). By comparing gene expression profiles in these KC subsets, we identified 26 additional genes whose expression levels significantly differ between them (*Figure 2F*). Of these, eleven are involved in gene regulation (*pros, Eip93F, Bx, bol, trv, Mef2, mamo, ab, Rbp6, Imp, fru*), five in signal transduction (*Ca-alpha1T, Pka-C3, CG8641, cmpy, norpA*), and three in synapse function (*Sytalpha, cmpy, DAT*), indicating plausible mechanistic differences between these three major classes of KCs. We independently validated these differential expression patterns by purifying mRNA from the three KC subtypes labeled with *mCherry* driven by the MB008B (αβ), MB131B (γ) or MB461B (α′β′) GAL4 drivers (*Aso et al., 2014*) and isolated by Fluorescence Activated Cell Sorting (FACS). Bulk mRNA was extracted from groups of 500–2500 of each KC type and real-time qPCR analysis was used to compare the expression levels for the 29 genes in *Figure 2F*. Despite the low starting amounts of mRNA, we obtained consistent qPCR signal for thirteen of these genes (*Figure 2—figure supplement 1*). Importantly, the differences in expression measured by qPCR for these 13 genes precisely matched the profiles detected in the Drop-seq data. These data confirm the accuracy of our measurements of expression with Drop-seq.

## Identification of olfactory projection neurons

We assigned two cell clusters containing PNs (*Figure 1B*), based on the strong expression of previously described markers, including *cut* (*ct*) and *abnormal chemosensory jump 6* (*acj6*). The *ct* gene encodes a homeobox transcription factor involved in dendrite targeting in PNs and is known to be expressed in a subset of the antero-dorsal (ad-), lateral (l-) and ventral (v-) PNs (*Komiyama and Luo, 2007*). The *acj6* gene encodes a POU-domain transcription factor that is also necessary for PN development and has been described to label all adPNs and a subset of lPNs (*Komiyama et al., 2003*; *Lai et al., 2008*) (*Figure 3C*). Although, other cells that are not PNs might express *ct* and *acj6* (*Certel et al., 2000*), these two putative PN clusters are the only ones strongly expressing, both of these genes. We therefore next focused analyses on these *acj6/ct* expressing cells and performed a new PCA and t-SNE analysis on the top six PCs. This segregated them into four distinct subclusters, each of which expresses a specific transcriptional signature (*Figure 3A–B*). Consistent with the expression patterns mentioned above, *ct* transcripts were found in all four clusters, whereas *acj6* was only identified in three (Clusters 1, 2 and 4; *Figure 3C*). Interestingly, *ventral veins lacking* (*vvl*), another POU-domain transcription factor reported to be expressed in *acj6*-negative lPNs (*Komiyama et al., 2003*; *Li et al., 2017*) only labeled a small number of neurons, which were all part of the cluster that was negative for *acj6* (Cluster 3; *Figure 3C*). Our data therefore confirm the non-overlapping expression patterns of *acj6* and *vvl*, and support the assignment of the *vvl* expressing cluster to cells including the lPNs.

To identify putative ventral PNs (vPNs), we used expression of *Lim1*, which encodes a LIM-homeodomain transcription factor reported to be expressed in most vPNs, but not in adPNs or lPNs (*Komiyama and Luo, 2007*; *Li et al., 2017*). Surprisingly, *Lim1* labeled one of the three *acj6*-positive clusters, and several neurons co-expressed both *Lim1* and *acj6* (Cluster 4; *Figure 3C*). This contrasts with a previous study that indicated that *acj6* and *Lim1* do not overlap, as a result of these two genes being expressed in progeny derived from discrete PN neuroblasts (*Komiyama and Luo, 2007*). About 50% of the *acj6*-positive neurons were recently shown to express *knot* (*kn*), another transcription factor involved in dendrite morphology (*Jinushi-Nakao et al., 2007*; *Li et al., 2017*).

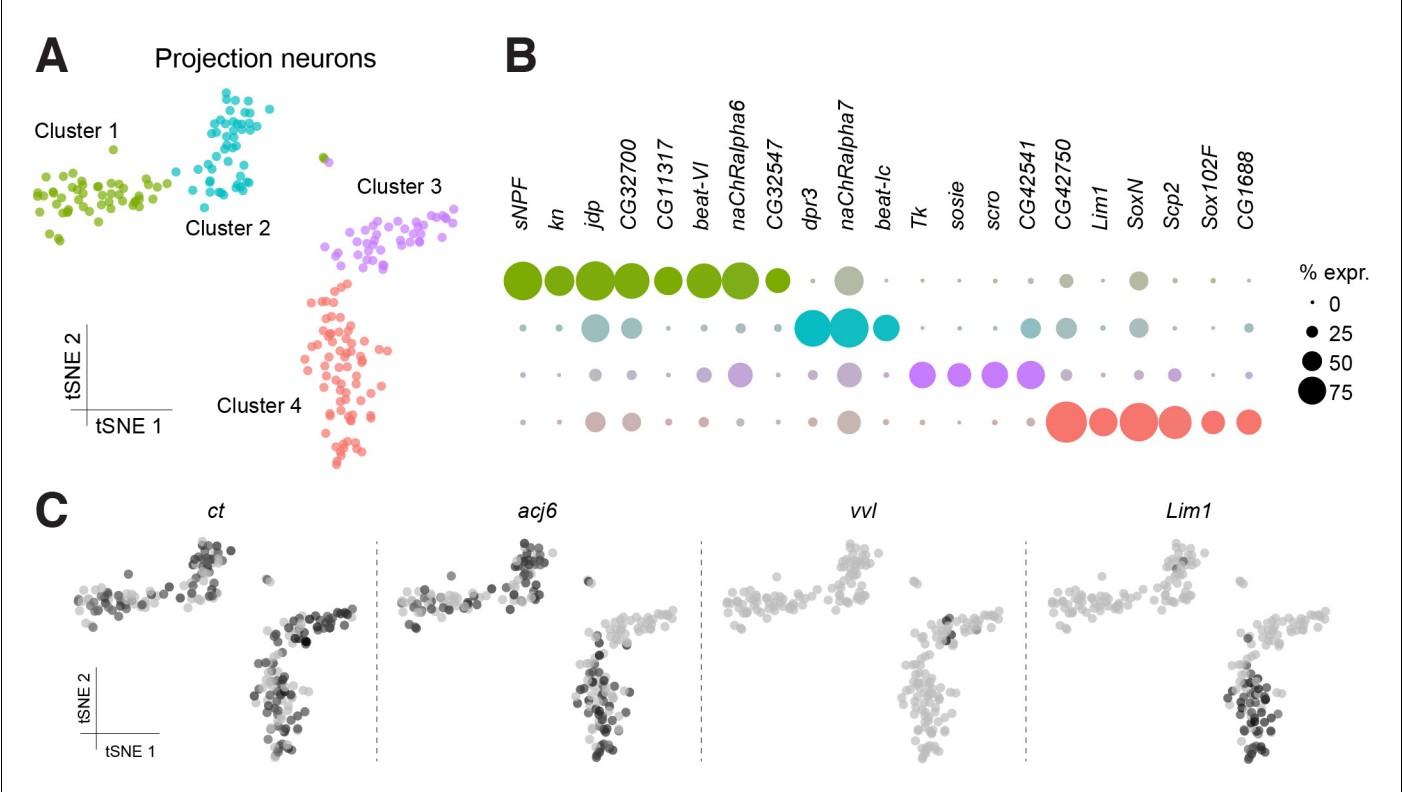

**Figure 3.** Sub-populations of olfactory projection neurons. (**A**) Re-clustering of the two Projection Neuron (PN) clusters from *Figure 1B*. Clusters are color-coded. (**B**) Dot plots showing the main markers distinguishing PN populations from each other (pairwise comparisons for genes expressed in >50% of cells in either cluster; Log2 FC >1.5, Wilcoxon rank-sum test with Bonferroni-corrected p-value<0.01). Dot diameter represents the fraction of cells expressing each gene in each cluster, as shown in scale. Color intensity represents the average normalized expression level. (**C**) t-SNE-plots for some known markers of PNs. *ct* labels all PN clusters, *acj6* and *vvl* are mutually exclusive in Clusters 1, 2 and 4, versus Cluster 3, while *Lim1* is specifically expressed in Cluster 4. Some of these cells may be antennal lobe local interneurons.

DOI: https://doi.org/10.7554/eLife.34550.012

Consistently, we found that the two *acj6*[+]/*Lim1*[-] clusters (clusters 1 and 2) segregate according to *kn* expression (*Figure 3B*).

We also identified three to eight genes in each PN cluster that were significantly over-expressed, as compared to the expression in other PN clusters (*Figure 3B*). Of potential functional importance, we found that the *acj6*[+]/*kn*[+] PNs strongly express the *sNPF* neuropeptide gene, whereas neurons encompassing the putative lPNs express *Tachykinin* (*Tk*). These data suggest that these two classes of otherwise cholinergic neurons might co-release different neuropeptides. Interestingly, the sNPF and Tk neuropeptides have previously been reported to have a modulatory role in the antennal lobe, although these studies concluded that the peptides were released from olfactory receptor neurons and local interneurons, respectively (*Ignell et al., 2009*; *Nässel et al., 2008*). More recently, others have also detected the expression of *Tk* in PNs (*Li et al., 2017*).

## Identification of glia and astrocytes

The two known neuronal markers *embryonic lethal abnormal vision* (*elav*) and *neuronal Synaptobrevin* (*nSyb*) (*DiAntonio et al., 1993*; *Robinow and White, 1988*) were broadly expressed in most cells but were conspicuously absent from four clusters (*Figure 4—figure supplement 1A–B*), indicating that these cell populations were possibly not neuronal. One of these clusters expressed a series of genes previously associated with fat body, such as *Secreted protein, acidic, cysteine-rich* (*SPARC*), *Metallothionein A* (*MtnA*), *I'm not dead yet* (*Indy*) and *pudgy* (*pdgy*) (*Catalán et al., 2016*; *Knauf et al., 2002*; *Shahab et al., 2015*; *Xu et al., 2012*) (*Figure 1—source data 2*). We therefore expect these cells to represent residual fat body tissue that remains after brain dissection. The three

other *elav/nSyb* negative clusters expressed two known glial markers: the homeobox transcription factor *reversed polarity* (*repo*) (*Xiong et al., 1994*) was found at variable levels, while the Na⁺/K⁺ transporting ATPase *nervana 2* (*nrv2*) (*Sun and Salvaterra, 1995*) was broadly expressed in all cells in these clusters (*Figure 4—figure supplement 1C–D*). Interestingly, one of these three putative glial clusters also robustly expressed the *astrocytic leucine-rich repeat molecule* (*alrm*) and *wunen-2* (*wun2*) genes (*Figure 4—figure supplement 1E*, *Figure 1—source data 2*), which are known to be specifically expressed in astrocytes (*Doherty et al., 2009*; *Huang et al., 2015*). We therefore define these three clusters 'Glia 1' and 'Glia 2' and astrocytes..

Given the previous morphological subdivision of glia into cortex, neuropil, surface and astrocyte types (*Freeman and Doherty, 2006*), we also attempted to sub-cluster the glial cell populations, as described above for PNs. However, this analysis did not reveal additional obvious cluster substructure (*Figure 4A*), suggesting that at this sequencing depth and number of cells, these three glial populations may be fairly homogeneous. However, we identified a large number of genes that were differentially expressed between these three clusters (*Figure 4B*, *Figure 1—source data 2*).

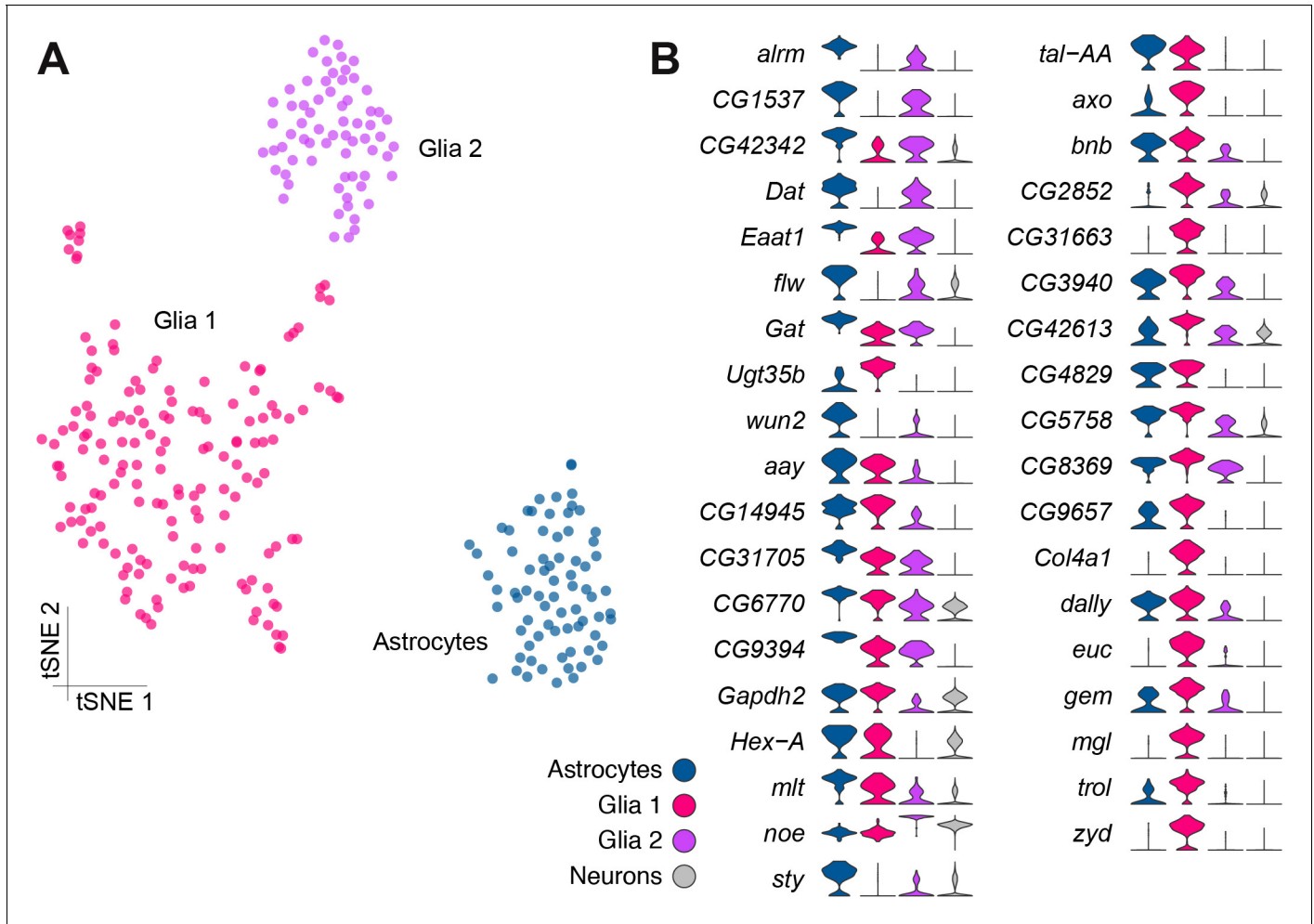

**Figure 4.** Genetic markers of glial subtypes. (A) Re-clustering of the glial and astrocyte populations identified in *Figure 1B*. The same three sub-clusters are identified. (B) Violin plots showing the main markers that distinguish glial subtypes from each other (pairwise comparisons for genes expressed in >75% of cells in either cluster; Log2 FC >2, Wilcoxon rank-sum test with Bonferroni-corrected p-value<0.01). The column on the right (grey) indicates the level of expression of each gene across all neurons in the brain (excluding glia, astrocytes and fat body).

DOI: https://doi.org/10.7554/eLife.34550.013

The following figure supplement is available for figure 4:

**Figure supplement 1.** Expression levels of glia-specific genes and Ilp 6.

DOI: https://doi.org/10.7554/eLife.34550.014

Importantly, several genes that are known to be glial- or astrocyte-specific are amongst the 37 genes we found with Drop-seq to be most differentially expressed between these glial clusters (*DeSalvo et al., 2014*; *Huang et al., 2015*). Unfortunately, these known markers do not permit us to assign Glial one and Glial two to a particular glial cell-type at this stage.

Surprisingly, cells in Cluster M robustly express both the neuronal markers *elav* and *nSyb* and also the glial *nrv2* and a number of other glial markers, although for the majority of them, in lesser amounts than in those clusters we assigned to glia above (*Figure 1—source data 2*). Since these cells are also notable for not expressing *repo* (*Figure 4—figure supplement 1D*), it seems plausible that they represent a novel hybrid cell type. However, we cannot exclude that they arose from fragments of cortex glia that remained attached to neuronal cell bodies.

## Assigning fast-acting neurotransmitters

We next assessed the proportion and distribution of cells in our data set that express genes that would indicate they release a particular fast-acting neurotransmitter; acetylcholine (ACh), glutamate (Glu) and gamma-aminobutyric acid (GABA). We determined that cells were cholinergic, glutamatergic or GABA-ergic based on the expression of *vesicular acetylcholine transporter* (*VAChT*), *vesicular glutamate transporter* (*VGlut*) and *glutamic acid decarboxylase 1* (*Gad1*), three key proteins that are either required for the vesicular loading, or metabolism, of ACh, Glu and GABA respectively. Consistent with our expectations, this analysis labelled the cell clusters that most likely represent KCs and PNs as being cholinergic (*Barnstedt et al., 2016*; *Tanaka et al., 2012*), while the ellipsoid body cluster is comprised of GABAergic cells (*Figure 5A*) (*Kahsai et al., 2012*). Reassuringly, we did not find significant neurotransmitter marker expression in glia, including astrocytes.

Cells expressing these neurotransmitter-specific marker genes were largely exclusive, although 8% of cells contained markers for ACh and GABA and 7% for ACh and Glu. It is therefore conceivable that some cells release excitatory and inhibitory neurotransmitters. A smaller percentage of cells expressed markers for Glu and GABA (3%), of which a third (1%) expressed all three neurotransmitter markers (although these possibly represent multiple cell captures) (*Figure 5B*).

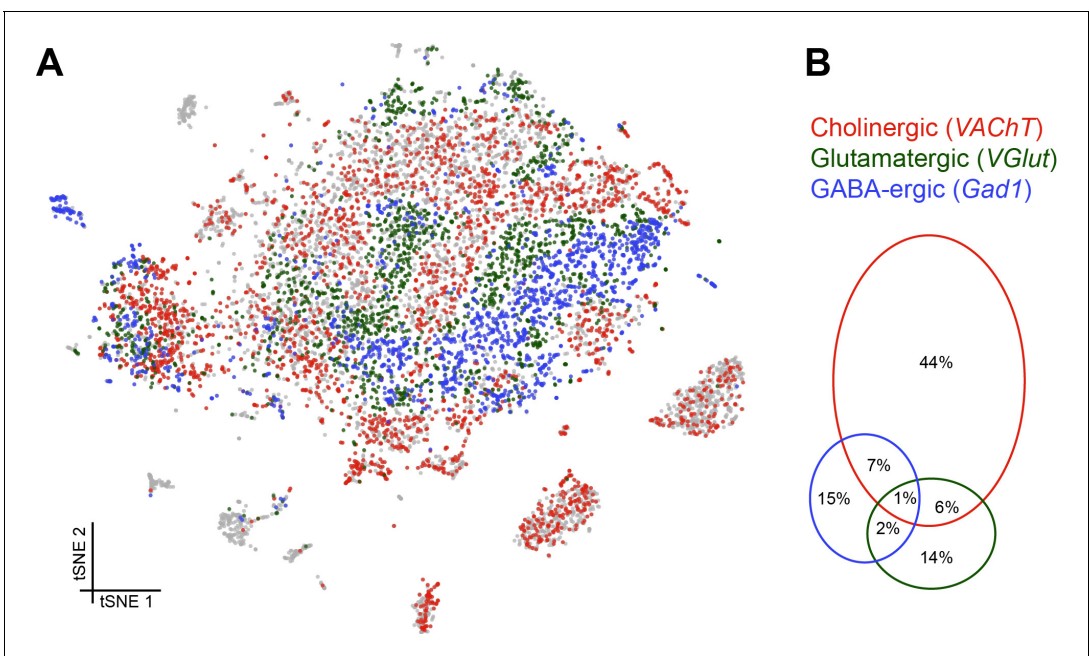

**Figure 5.** Distribution of fast-acting neurotransmitters. (**A**) t-SNE plot showing the distribution of cells expressing *vesicular acetylcholine transporter* (*VAChT*, cholinergic neurons), *vesicular glutamate transporter* (*VGlut*, glutamatergic neurons) and *glutamic acid decarboxylase 1* (*Gad1*, GABA-ergic neurons). For graphical reasons only cells expressing each marker above a log normalized value of 2 are shown. (**B**) Quantification of cells expressing markers displayed in A. The difference to 100% are cells that did not express any of the three markers.
DOI: https://doi.org/10.7554/eLife.34550.015

## Analysis of neuropeptide expression

We also analyzed the expression of neuropeptides in our Drop-seq dataset. We first investigated whether individual neuropeptide-encoding genes were preferentially expressed in neurons that co-transmit/co-release a particular fast-acting neurotransmitter ACh, Glu or GABA (*Figure 6A*). *sNPF*, *CCHamide-2* (*CCHa2*), *Tk*, *space blanket* (*spab*), *jelly belly* (*jeb*) and *amnesiac* (*amn*) showed a strong preference for expression in cholinergic neurons, whereas *Diuretic hormone 31* (*Dh31*) is highly biased to GABA-ergic neurons. *Neuropeptide-like precursor 1* (*Nplp1*) and *Allatostatin A* (*AstA*) were mainly expressed in glutamatergic cells.

Some other peptide-encoding genes show a strong anti-correlation with a particular transmitter. For example *neuropeptide F* (*dNPF*), *sNPF*, *Tk*, *spab*, *jeb*, *Allatostatin C* (*AstC*), *Diuretic hormone 44* (*Dh44*), *CCHa2* and *Myosuppressin* (*Ms*) were anti-correlated with GABA-ergic cells. Similarly, *Myoin-hibitory peptide precursor* (*Mip*), *pigment-dispersing factor* (*PDF*) and *SIFamide* (*SIFa*) were absent from cholinergic neurons.

*Ms* showed an interesting bias for expression in cells that express two (Glu and ACh or Glu and GABA) or all three fast acting neurotransmitters. We also noticed that the specificity towards cells expressing only one type of fast-acting neurotransmitter varied between neuropeptides, with some such as PDF, exhibiting a broad and general expression pattern, other than the anti-correlation with ACh.

The abundance and specificity of expression across the midbrain also varied between individual neuropeptides. Some neuropeptide-encoding genes are only expressed in 1–2% of cells (e.g. *CCHa2*, *amn*, *dNPF*, *Mip*, *PDF* and *SIFa*), and their release could therefore potentially represent signals of, for example, internal states. Others, such as *spab*, *sNPF* and *Nplp1*, are very broadly expressed in 20–25% of all cells (see *Figure 1—source data 1*), suggesting that these neuropeptides likely act as modulatory co-transmitters with fast-acting neurotransmitters.

Some neuropeptide expression patterns are highly specific to certain cell types. For example, *Dh31* is mainly expressed by ellipsoid body neurons whereas *sNPF* is strongly expressed in αβ and γ KCs (*Figure 2E*), in $acj6^+/kn^+$ PNs (*Figure 3B*) and in clusters C and D, that have not yet been assigned to a specific cell-type. Furthermore, although both *spab* and *Nplp1* are very broadly expressed, their expression patterns are strongly anti-correlated, suggesting that they may have complementary functions in the *Drosophila* midbrain.

We also found transcripts for the *Drosophila* insulin-like peptides 2, 3, 5 and 6 (*Figure 6B*). The Ilp2, Ilp3 and Ilp5 peptides are exclusively expressed in IPCs in the brain, whilst Ilp6 is expressed in glia (*Brogiolo et al., 2001*; *Okamoto et al., 2009*). We found that Ilp2, 3 and 5 expression was only weakly correlated with that of neurotransmitters, whilst Ilp6 expression is strongly correlated with cells that do not express neurotransmitter markers, but that are positive for the glia-specific genes *repo* and *nrv2* (*Figure 4—figure supplement 1*) (*Freeman et al., 2003*; *Sun and Salvaterra, 1995*; *Xiong et al., 1994*).

## Assignment and subdivision of monoaminergic neurons

We used expression of the *vesicular monoamine transporter* (*Vmat*) gene to identify monoaminergic neurons in our midbrain dataset (*Figure 7A*). Three discrete cell populations clearly expressed *Vmat*. We performed a new PCA and tSNE analysis on cells from these three clusters, guided by known markers for serotonin (5-HT), tyramine (TA), octopamine (OA) and dopamine (DA) releasing neurons. *Dopa decarboxylase* (*Ddc*) labels 5-HT and DA neurons, *Serotonin transporter* (*SerT*) and *Tryptophan hydroxylase* (*Trh*) mark 5-HT neurons, *pale* (*ple*; tyrosine hydroxylase) and *Dopamine transporter* (*DAT*) DA neurons, *Tyrosine decarboxylase 2* (*Tdc2*) TA and OA neurons, *Tyramine β-hydroxylase* (*Tbh*) OA neurons. These labels allowed us to identify the neuronal clusters corresponding to each of these cell types (*Figure 7B*).

In addition to known markers of monoaminergic neuronal types, we found new genes expressed in these populations (*Figure 7C*), that are likely to have an important role for their development and connectivity, such as *kekkon 1* (*kek1*) in dopaminergic neurons (DANs) (*Ghiglione et al., 1999*), or *IGF-II mRNA-binding protein* (*Imp*) and *Jim Lovell* (*lov*) in serotonergic neurons (*Bjorum et al., 2013*; *Geng and Macdonald, 2006*; *Munro et al., 2006*). High expression in TA neurons of *hikaru genki* (*hig*), which encodes a protein generally found in the synaptic clefts of cholinergic synapses (*Nakayama et al., 2014*; *2016*), may highlight the importance of cholinergic input to these neurons.

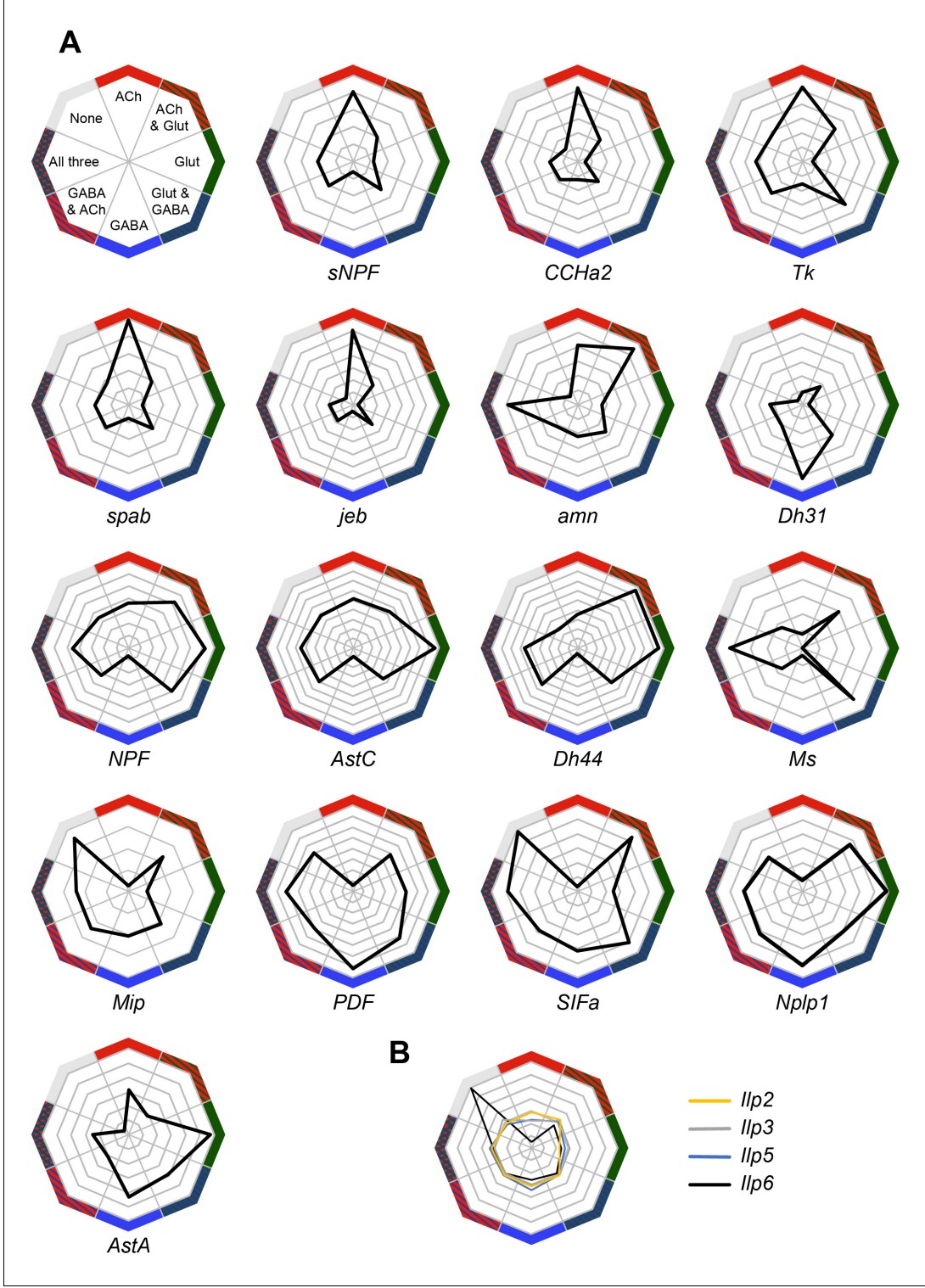

**Figure 6.** Co-expression of neuropeptides with fast-acting neurotransmitters. (**A**) Radar plots showing the co-expression of 16 neuropeptides with the three fast-acting neurotransmitters. Data represents the relative abundance of cells expressing a neuropeptide and either VAChT (ACh), VGlut (Glut), Gad1 (GABA), combinations of these three genes, or none of them. (**B**) Co-expression of four *insulin-like peptides*, including the non-neuronal *Ilp6*, with fast-acting neurotransmitters.

DOI: https://doi.org/10.7554/eLife.34550.016

Many neurons release more than one neurotransmitter. We therefore investigated whether the different types of monoaminergic neurons co-expressed markers for fast-acting transmitters. The most striking evidence in these analyses suggest that many octopaminergic and tyraminergic neurons likely co-release Glu, and less of them GABA, or ACh. (*Figure 7D*).

We also tested whether monoaminergic neurons co-expressed neuropeptide genes. Many mature neuropeptides are amidated at their C-terminus through the sequential enzymatic action of the *Phm*- and *Pal2*-encoded peptidylglycine-alpha-hydroxylating monooxygenase and peptidyl-alpha-hydroxyglycine alpha-amidating lyase (*Han et al., 2004*; *Jiang et al., 2000*; *Kolhekar et al., 1997*). These genes were expressed in 50% and 81% of all monoaminergic neurons, respectively (*Figure 7E*), suggesting that a significant proportion of monoaminergic neurons likely co-release neuropeptides. Indeed, we found expression of *Dh44*, *Nplp1*, *Glycoprotein hormone beta 5* (*Gpb5*) and *Proctolin* (*proc*; which is not amidated) in up to 21% of DANs (*Figure 7E*). 61% of DANs express at least one neuropeptide and 32% express two, or more. *Dh44*, *Nplp1* and *spab* were found in up to 30% of 5-HT neurons, with 90% of these expressing one or two neuropeptides (*Figure 7E*).

Perhaps surprisingly, OA and TA neurons contained mRNA for many neuropeptides. We found that 85% of OA neurons express at least one neuropeptide, whereas 46% express two, or more. Co-expression was even more evident in TA neurons; 83% expressed one, whereas 78% express two or more. *Nplp1*, *Gpb5*, and *SIFa* were detected in TA and OA neurons, whereas *Dh44*, *Ms* and *spab* were only identified in OA neurons, and *sNPF*, *Dh31*, *Mip*, *Ilp2* and *ITP* were exclusively found in TA neurons (*Figure 7E*). *Dh44* was the most broadly expressed, being detected in 46% of OA neurons. *Mip* and *SIFa* were each expressed in 44% of TA neurons, and were co-expressed in 33% of them. Together, these results indicate that neuropeptide expression, and co-expression, is a common feature of many monoaminergic neurons. The obvious complexity and possible heterogeneity of neuropeptide expression may reflect functional specialization of individual, or small groups of these monoaminergic neurons.

Prior work has shown that DANs are anatomically and functionally divisible based on roles in motivation, learning and memory and arousal (*Huetteroth et al., 2015*; *Krashes et al., 2009*; *Nall et al., 2016*; *Yamagata et al., 2015*). Some of this DAN subdivision has also been associated with the expression of particular transcription factors, receptors for specific neuropeptides, or other monoamines (*Bou Dib et al., 2014*; *Ichinose et al., 2015*; *Krashes et al., 2009*). DANs implicated in learning and memory reside in two discrete clusters, called PPL1 and PAM. PPL1 DANs mostly convey the negative reinforcing effects of aversive stimuli, such as electric shock, high heat or bitter taste (*Aso et al., 2012*; *Das et al., 2014*; *Galili et al., 2014*), whereas the numerically larger PAM cluster contains DANs that appear somewhat specialized in representing particular types of rewards, such as the sweet taste and nutrient value of sugars, or water (*Burke et al., 2012*; *Huetteroth et al., 2015*; *Lin et al., 2014*; *Liu et al., 2012*; *Yamagata et al., 2015*). Prior work demonstrated that PAM DANs express the transcription factor *48 related 2* (*Fer2*), which is required for their development and survival (*Bou Dib et al., 2014*). We found that 44 neurons in the DA cluster (37%) express *Fer2* (*Figure 7B*). We therefore consider these *Fer2*-positive cells to represent PAM DANs. 15 additional genes are significantly over-expressed in these cells, in comparison to the rest of the brain (*Figure 7F*). Amongst them we found *Ddc*, *ple*, *Vmat* and *DAT*, that are essential for DA synthesis, vesicle loading and transport (*Yamamoto and Seto, 2014*). Potential new markers for PAM DANs include the transcription factor *scarecrow* (*scro*), the amino-acid transporter *Jhl1-21*, the Dpr-interacting protein *DIP-delta*, the PDGF- and VEGF-related growth factor *Pvf3*, the EGFR modulator *kek1*, as well as five novel genes; *CG1402*, *CG13330*, *CG17193*, *CG10384* and *CG42817*.

To corroborate the expression of these new markers in PAM neurons, we compared this data to a transcriptome profiling dataset that we acquired from sequencing mRNA extracted from populations of GFP labeled PAM DANs. We used R58E02-GAL4, a PAM-specific line (*Liu et al., 2012*; *Pfeiffer et al., 2008*) to express UAS-6xGFP (*Shearin et al., 2014*) specifically in PAM DANs, and purified the cells by FACS. We prepared mRNA from GFP + and GFP- neurons, which was subsequently reverse-transcribed and amplified using Smart-seq2, and sequenced. This analysis identified about 10 times more (143) genes that were significantly over-expressed in PAM neurons, as compared to the number retrieved with Drop-seq (*Figure 7F*). This return is consistent with previous reports showing a higher recovery rate with Smart-seq2 compared to Drop-seq, but also higher levels of noise, as Smart-seq2 does not employ UMIs (*Ziegenhain et al., 2017*). Of the 15 genes found to be over-expressed in PAM neurons in the Drop-seq experiment, 9 (*ple*, *DAT*, *Fer2*, *Jhl-21*, *scro*,

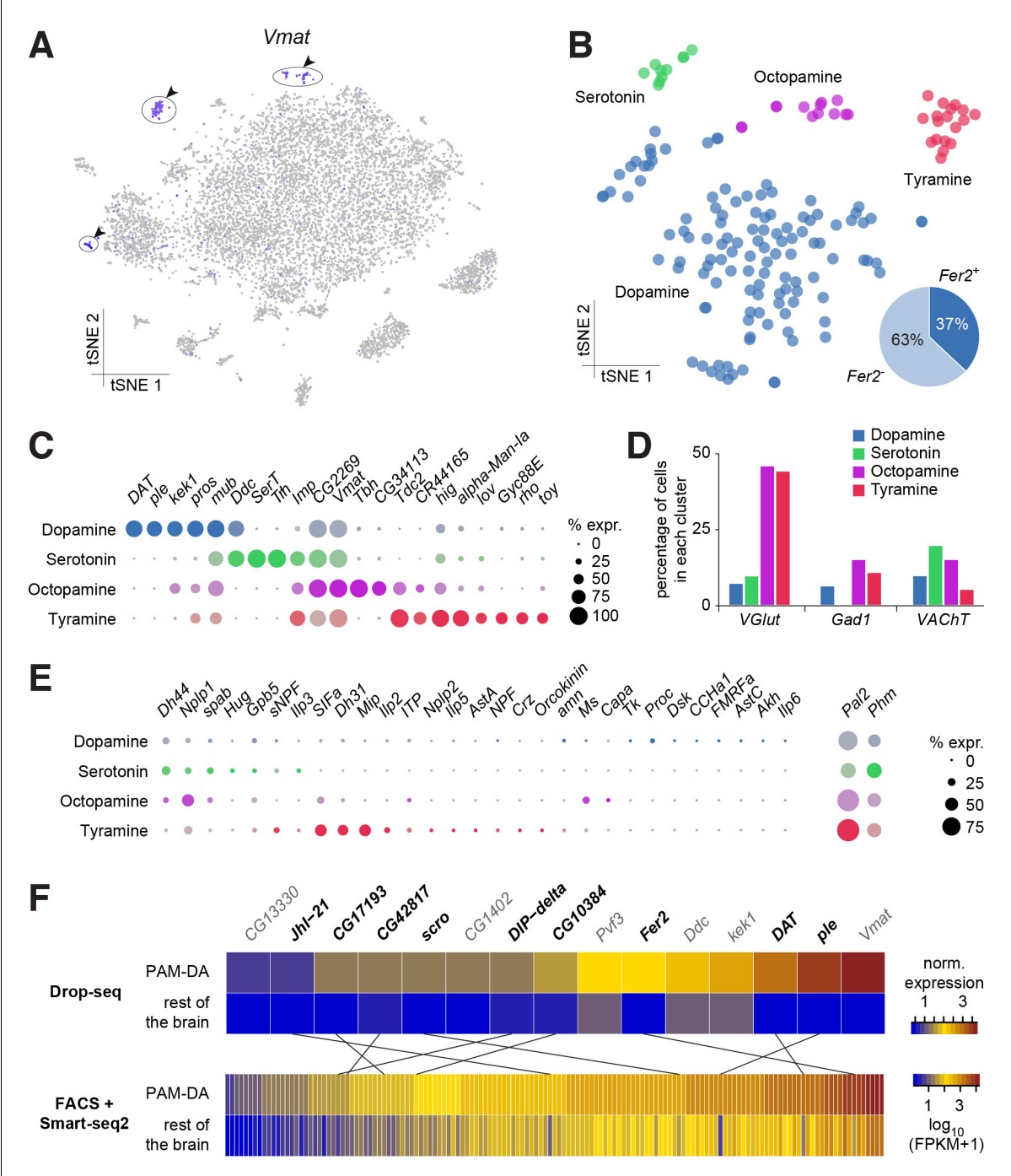

**Figure 7.** Genetic markers and co-transmission in monoaminergic neurons. (**A**) Expression of the Vesicular Monoamine Transporter (*Vmat*) labels three clusters from *Figure 1B* (cells in indigo, highlighted with arrowheads and circles. (**B**) Re-clustering of the three cell populations labeled in (**A**). Four sub-clusters are identified, representing dopaminergic, tyraminergic, octopaminergic, and serotonergic neurons. (**C**) Dot plots showing the main markers distinguishing monoamine populations from each other (pairwise comparisons for genes expressed in >50% of cells in either cluster; Log2 FC >1, Wilcoxon rank sum test with Bonferroni-corrected p-value<0.01). Dot diameter represents the fraction of cells expressing each gene in each cluster, as shown in scale. Color intensity represents the average normalized expression level. (**D**) Percentage of cells in each monoaminergic cluster that are co-expressing markers for fast neurotransmitter-releasing neurons (*VGlut*, *Gad1*, and *VAChT*). (**E**) Dot plots showing expression of genes encoding neuropeptides and neuropeptide amidating enzymes across monoaminergic populations. Dot diameter represents the fraction of cells expressing each gene in each cluster, as shown in scale. Color intensity represents the average normalized expression level. (**F**) Comparison of genes overexpressed in PAM dopaminergic neurons compared to the rest of the brain, measured with Drop-seq or with FACS and Smart-seq2. 9 of the 15 genes identified with Drop-seq (labeled in bold) were also found in the Smart-seq2 dataset.

DOI: https://doi.org/10.7554/eLife.34550.017

*DIP-delta*, *CG10384*, *CG17193* and *CG42817*) were also retrieved in the Smart-seq2 data (*Figure 7F*). This confirms that these genes are specifically expressed in PAM neurons. Furthermore, with the exception of *ple*, *DAT* and *Fer2*, these genes have not been previously localized to PAM neurons, and therefore represent novel markers for this cell-type.

## Dopamine receptors

Cells respond to DA using a variety of DA receptors in their cell membrane. Interestingly, our analysis shows that all four DA receptors (*Dop1R1, Dop1R2, Dop2R* and *DopEcR*) are found in KCs, which form numerous synapses with DA neurons in the mushroom body lobes (see above; *Figure 8B*). However, our analyses suggest that *Dop2R* is less abundant in KCs than the three other receptors. Many KCs appear to express multiple DA receptors but only 24% (250 of 1041) were found to co-express all four (*Figure 8C*). The data also suggest that,individual αβ KCs may express a combination of fewer types of DA receptors, than do α′β′ and γ KCs (*Figure 8C*). However, since we found 4.7% of KCs (49 of 1041) that do not express any DA receptors, we cannot exclude that low-level expression accounts for some of these apparent differences,

Other cell types also express combinations of DA receptors, to varying degrees. In addition to KCs, *Dop1R1* (*dumb*) and *Dop1R2* (*damb*) are found in a few other clusters (*Figure 8B*), in particular in several of those that we could not attribute to any cell type. This information will be helpful for further characterizing these clusters. Consistent with evidence showing that *Dop1R1* is expressed in the Central complex where it regulates arousal (*Kahsai et al., 2012*; *Lebestky et al., 2009*), we found limited expression of *Dop1R1* in the ellipsoid body,suggesting that only a subset of these neurons are involved in this process. *Dop1R1* also seems to be expressed in small numbers of monoaminergic neurons, suggesting that it may play a role in autocrine signaling. However, the main candidate receptor for DA autocrine signaling is *Dop2R*, which was found to be broadly expressed in DANs, and also in large numbers of other monoaminergic neuronal types (*Figure 8B*). Interestingly, *Dop2R* expression was also detected in some PNs and IPCs as well as a few non-attributed clusters, which indicates that the activity of these neurons is also subject to dopaminergic modulation. Finally, the Dopamine/Ecdysteroid receptor (*DopEcR*) was found in several cell types, including KCs, PNs, the ocelli, and many other non-attributed clusters (*Figure 8B*), suggesting a broad role for this receptor. Expression of this *DopEcR* in PNs corroborates previous data showing its involvement in pheromone sensitization in these neurons, both in flies and moths (*Abrieux et al., 2014*; *Aranda et al., 2017*).

## Dopamine metabolism

DA signaling is regulated by enzymatic degradation and reuptake through transporters. Recycled metabolites can then be used to resynthesize DA. These steps can occur in different cell types, that could be DA-releasing cells, post-synaptic neurons, or glia (*Yamamoto and Seto, 2014*) (*Figure 8A*). We therefore used our Drop-seq data to determine which cell types expressed components of the DA recycling and metabolic pathways.

As expected, the first step of DA synthesis, conversion of tyrosine into the DA precursor L-DOPA catalyzed by the *ple*-encoded Tyrosine hydroxylase appears to occur exclusively in DANs (*Figure 8B*). In comparison *Ddc*, which converts L-DOPA to DA, is also involved in 5-HT synthesis, and so was expressed in DA and 5-HT neurons. Interestingly, *Ddc* also labels several other neuronal populations, including α′β′ and γ KCs, one cluster of olfactory PNs, and several non-identified, alphabet labeled clusters (*Figure 8B*). It is not clear if Ddc in these neurons is involved in the metabolism of DA or other aromatic L-amino acids.

Three enzymes have been described to play a role in DA degradation and recycling. The *ebony* (*e*) gene product converts DA into N-beta-alanyldopamine (NBAD) (*Hovemann et al., 1998*; *Suh and Jackson, 2007*) and was almost exclusively expressed in astrocytes in our data (*Figure 8B*). Dopamine-N-acetyltransferase, encoded by *Dat*, converts DA into N-acetyl *dopamine* (NADA). Interestingly, *Dat* was abundant in astrocytes, in smaller amounts in other glia, and was also detected in the ellipsoid body and a few other subsets of neurons (*Figure 8B*). Although these results highlight the important role of glia, and in particular astrocytes, in DA reuptake, metabolism and recycling, other cells appear to convert DA into NADA rather than into NBAD. The fate and consequence of these two metabolites in each cell type remains largely unknown. Finally, *tan* (*t*), a gene coding for a hydrolase that can convert NBAD back into DA, was not found in any cell population from the

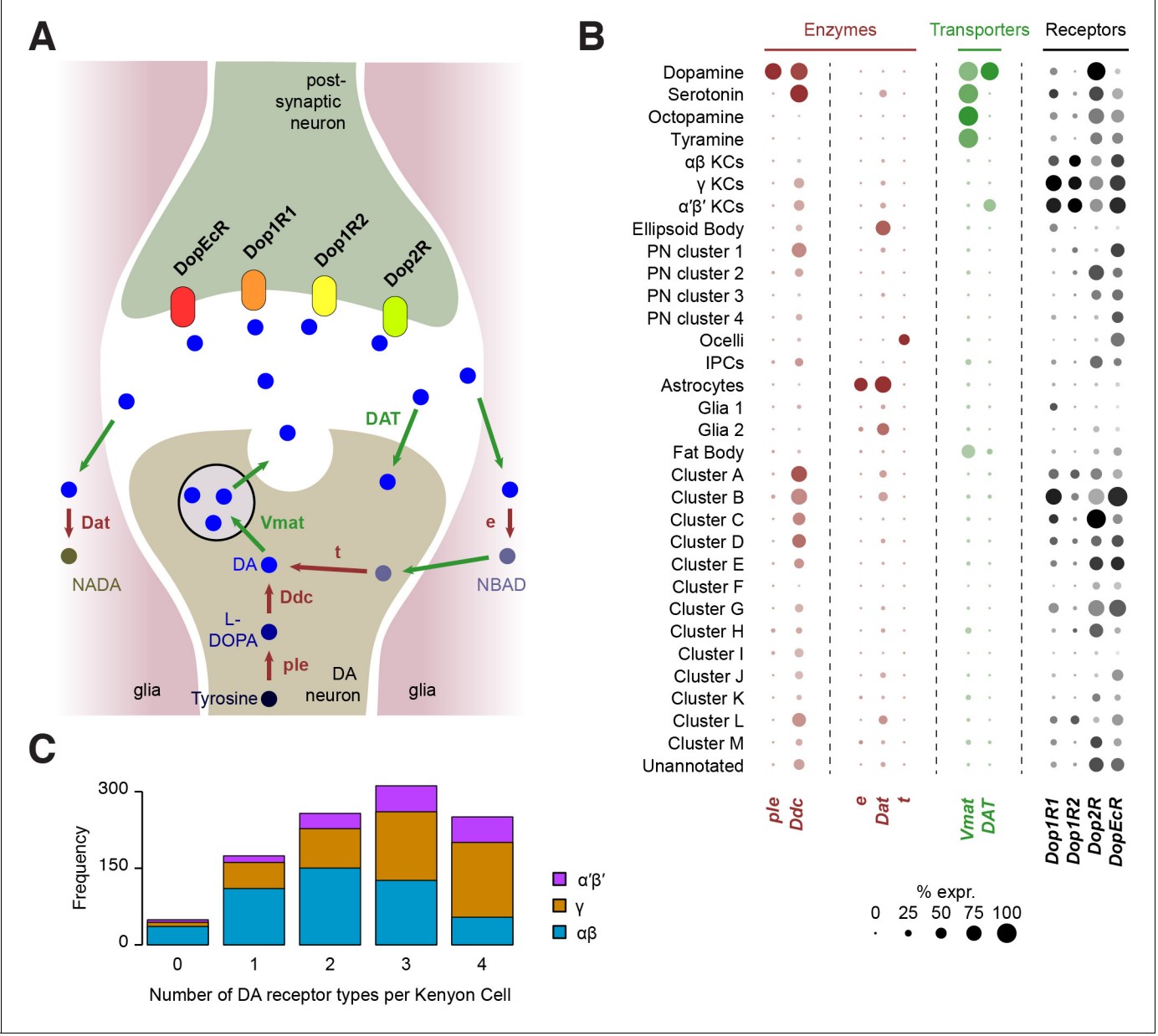

**Figure 8.** Localization of genes involved in dopamine metabolism and signaling. (A) Schematic of a dopaminergic synapse, representing the major proteins involved in dopamine signaling and metabolism. Drawing inspired by *Yamamoto and Seto (2014)* (B) Dot plots showing the expression of these genes across all cell populations identified in the *Drosophila* brain. Dot diameter represents the fraction of cells expressing each gene in each cluster, as shown in scale. Color intensity represents the average normalized expression level. (C) Histogram showing the number of DA receptor types in Kenyon Cells (as labeled in *Figure 2C & D*). Most cells express a combination of several receptor types.

DOI: https://doi.org/10.7554/eLife.34550.018

central brain itself (*Figure 8B*), suggesting that this recycling pathway is not utilized there. However, several cells from the ocelli express this enzyme, consistent with the function of *t* in histamine metabolism in photoreceptors (*Borycz et al., 2002*; *True et al., 2005*).

The vesicular monoamine transporter (encoded by *Vmat*) transports DA, 5-HT, OA and TA into synaptic vesicles (*Martin and Krantz, 2014*). As already mentioned, *Vmat* was detected in all these neuronal types. In addition, *Vmat* expression was evident in fat body cells. Although this has, to our knowledge, never been demonstrated in flies, perivascular adipose tissue in rats contains monoamines acting on the sympathetic nervous system, and is thus likely to express VMAT (*Ayala-*

*Lopez et al., 2014*). The *DAT*-encoded DA transporter mediates DA reuptake by DANs. Unlike *Vmat*, *DAT* was specifically expressed in dopaminergic, but not other monoaminergic neurons. Surprisingly, we also found *DAT* expression in α'β' KCs, suggesting that these neurons might tightly regulate the duration and magnitude of the DA signals that they receive.

## Distribution of nicotinic neurotransmitter receptors

The response of a neuron to a particular neurotransmitter is determined by the types of receptors that that cell expresses. In addition, most ionotropic neurotransmitter receptors are oligomers comprised of combinations of subunits, variations of which can have very different functional characteristics (*Sattelle et al., 2005*). Acetylcholine is a major excitatory neurotransmitter in the insect brain and is the primary fast-acting neurotransmitter released from olfactory receptor neurons, olfactory PNs and mushroom body KCs. Nicotinic acetylcholine receptors (nAChR) are heteropentamers that can be comprised of 2 or three alpha and the corresponding 3 or two beta subunits. Flies have seven alpha subunit genes and 3 types of beta encoding genes. These receptors have mostly been studied at the vertebrate neuromuscular junction (*Albuquerque et al., 2009*) and very little is known about the composition of nAChR in neurons in a central nervous system. Although gene expression cannot explicitly inform of subunit composition, co-expression is a prerequisite that limits the potential complexity in any given neuron. We therefore analyzed the co-expression of nAChR subunits using all cells from our Drop-seq dataset. We detected the expression, at varying frequencies, of all seven known nAChR α-subunits, and two of the three known β-subunits in our samples (*Figure 9A*). α1, α5, α6 and α7 are expressed in considerably more cells than α2, α3 and α4, whereas β1 is expressed in more than twice as many cells as β2. Most subunits are broadly expressed across all cell types, although some exhibit very distinct expression patterns. Most notably, α3 is broadly expressed in the midbrain, but distinctly absent in KCs. We also tested for co-expression of different combinations of receptor subunits (*Figure 9B*). Expression of α5 most strongly correlated with expression of α6 and β1. In contrast α3 weakly correlated with expression of α6 and β2 and α2 weakly with α4. Some of the patterns of expression are consistent with previously published pharmacological studies that tested for co-assembly of receptors by co-immunoprecipitation using α-Bungarotoxin (*Chamaon et al., 2002*; *Schulz et al., 2000*). For example, cells that express *nAChR-α1* most frequently also express *α2*, when compared to all other nAChR subunits and these two subunits have been shown to preferentially co-assemble into the same receptor complex (*Chamaon et al., 2002*; *Schulz et al., 2000*). Similarly, *β1* is the most frequently co-expressed subunit in *β2* expressing cells, again confirming previous co-immunoprecipitation experiments. We also detected high expression levels of the secreted protein *quiver* (*qvr*), a Ly-6/neurotoxin family member, in most neurons of our

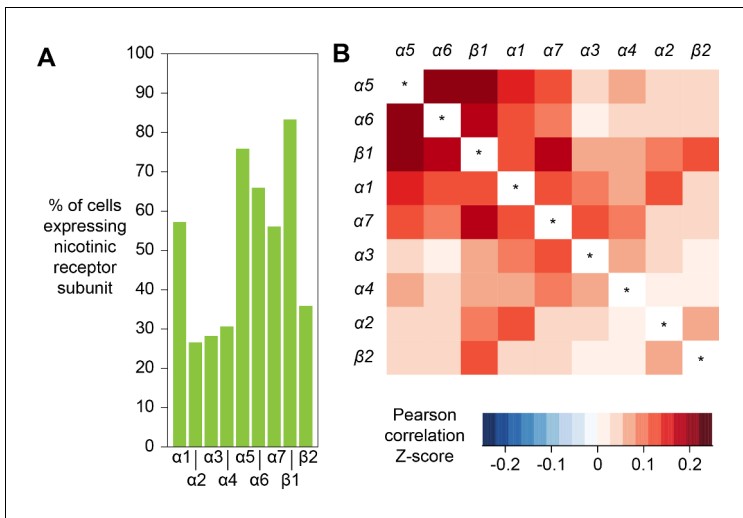

**Figure 9.** Expression patterns of nicotinic acetylcholine receptor subunits. (**A**) Prevalence of nicotinic receptor subunits. (**B**) Heatmap showing Pearson correlation coefficient Z-scores for each receptor subunit pair.
DOI: https://doi.org/10.7554/eLife.34550.019

sample, with no preference for neurons using a particular transmitter (see *Figure 1—source data 1*). The mammalian homologue of quiver, lynx1, has been shown to bind and regulate nAChR in the mammalian nervous system (*Miwa et al., 1999*).

## Co-expression of activity regulated genes

A recent study identified a set of genes whose expression was upregulated in response to prolonged neuronal activation. These activity-regulated genes were identified using differential bulk transcription profiling following broad neuronal activation, using three different artificial stimulation paradigms (*Chen et al., 2016*). We plotted the expression patterns of the 11 most highly upregulated genes that were identified following pan-neuronal optogenetic neuronal activation and found that 10 of them were also robustly expressed in our dataset (see *Figure 1—source data 1*). Interestingly, the expression patterns of the most highly upregulated activity-regulated genes were strongly correlated (*Figure 10*). For example, cells that express the transcription factor *stripe* (*sr*) are more likely to also express *Hormone receptor-like in 38* (*Hr38*, p-value$<2.2\times10^{-16}$, Pearson's product-moment correlation) and *CG14186* (p-value$<2.2\times10^{-16}$). These three genes were the most highly upregulated in *Chen et al., 2016*, following artificial optogenetic neuronal stimulation. Our data therefore demonstrate that they are also likely to be co-regulated in the brain, following ordinary levels of neuronal activity.

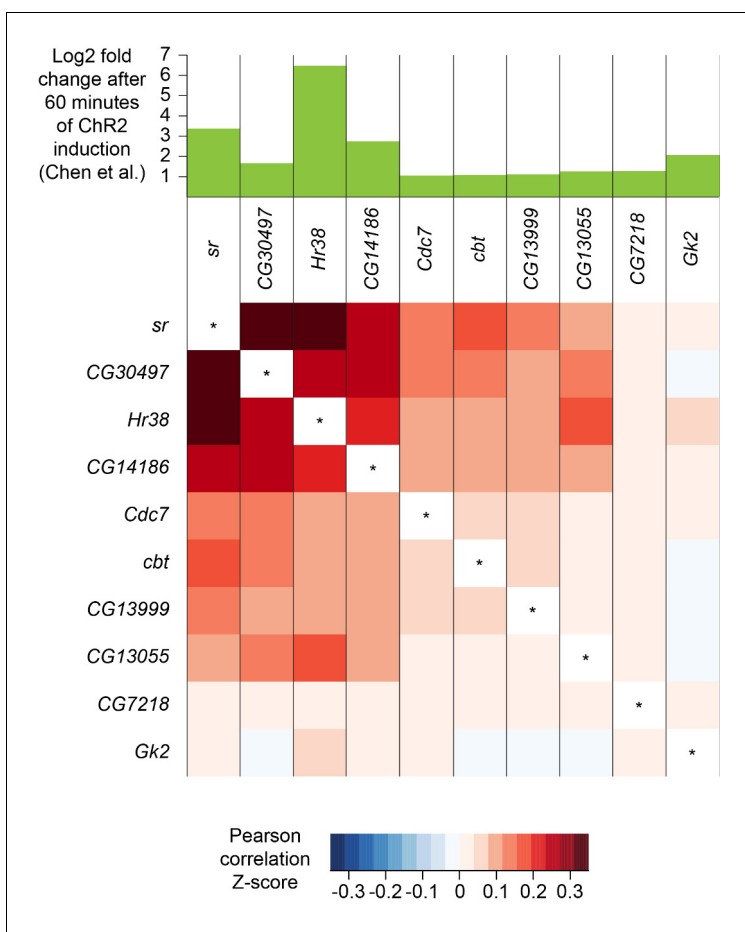

**Figure 10.** Co-expression of neuronal activity markers. Heatmap showing Pearson correlation coefficient Z-scores of activity-regulated genes, as reported by *Chen et al., 2016* (histogram on top). Ten most highly upregulated genes following ChR2-XXL-induced activation of all neurons that are expressed in the brain, ranked by their correlation.

DOI: https://doi.org/10.7554/eLife.34550.020

We wondered whether the expression pattern of these activity-regulated genes might highlight areas of the fly midbrain that have a high intrinsic level of activity. However, no specific cluster was prominently marked with the expression of 9 out of the 10 activity-regulated genes tested. Only CG13055 strongly labeled the cluster of γ KCs (*Figure 2F* and *Figure 1—source data 2*). We also noticed that expression of most activity-regulated genes was slightly higher in γ KCs. Since prior work suggested that the γ neurons are the least active of all the KC subtypes (*Turner et al., 2008*), we speculate that activity-regulated gene expression might be part of a homeostatic neuronal response to reduce excitability.

## Discussion

Generating an atlas of gene expression of every cell type in the human body is a goal of modern science (*Regev et al., 2017*). Remarkable recent advance in high-throughput single-cell RNA sequencing methods have brought this ambitious goal within reach. However, the large size of mammalian tissues means that huge numbers of cells need to be sequenced in order to capture a representative sample of the overall number. Insects, such as *Drosophila*, provide an obvious solution to the tissue size and cell number issues. Flies are complex organisms with tissues that serve analogous functions to many mammalian organs. Moreover, each one of these fly tissues is comprised of a greatly reduced number of cells, compared to their mammalian equivalent. This is perhaps most obvious when considering the brain. Whereas the mouse brain is estimated to contain about 75 million neurons, the *Drosophila* brain has only 150,000. Since two thirds of these cells comprise the optic lobes, much of the computational cognitive power and behavioral orchestration is handled by about 50,000 neurons in the midbrain. In this study, we describe a global and unbiased single-cell transcriptomic analysis, using Drop-seq (*Macosko et al., 2015*), that is representative of much of the *Drosophila* midbrain. This initial cell atlas of the fly brain provides a unique resource of gene expression across many cell types and regions of neuropil.

The extent of neural diversity is not known in any brain. Analysis of the fly therefore provides a useful inroad to this question. Even our initial clustering indicates a high level of neural complexity in the fly brain. Labeling the cluster plot with markers for the ACh, Glu and GABA neurotransmitters reveals that many diverse cells use each of these fast-acting neurotransmitters. For example, although the three major subsets of mushroom body KCs are all cholinergic they each occupy a discrete cluster and are distant to many other cholinergic neurons. The GABA-ergic ring neurons of the ellipsoid body are similarly unique and distinct from other GABA-ergic neurons. At this stage, we cannot tell whether cells in the major KC subtype and ellipsoid body neuron clusters are truly homogenous, or whether further iterative clustering will separate them into additional distinguishable subtypes. We might expect to find that the anatomically unique core, surface and posterior subdivisions of the αβ KCs have unique molecular profiles within the larger αβ cluster (*Aso et al., 2014*; *Hattori et al., 2017*; *Ito et al., 1997*; *Lee et al., 1999*; *Lin et al., 2007*; *Perisse et al., 2013*; *Tanaka et al., 2008*). Similarly, the ellipsoid body ring neurons might be separable into layer specific subtypes (*Wolff et al., 2015*). This will require additional analyses and perhaps the collection of more cells. Comparing Drop-seq profiles from the fly brain to those from larger social insects, such as ants and honeybees, and to neurons from the mammalian brain would be useful to address the question of how a larger brain is constructed. As a brain evolves to be bigger, are there many more cell types? Or is there simply an expansion of the number of copies of each cell-type? One might imagine that just expanding the number of identical cortical units, such as pyramidal neurons or mushroom body KCs, increases the computational power of the brain by permitting a higher degree of parallel processing and that the resulting larger networks also provide more storage space.

A key element of our analysis here is the ability to assign many single-cell molecular signatures to the relevant cell-type and brain region. We did this using a number of different approaches. Our data was collected from individual neurons taken from brains that specifically express mCherry in the αβ KCs of the mushroom body. This allowed us to unequivocally identify these neurons in our cluster plot and demonstrates the power of sequencing cells from a brain where some specific neurons are genetically labeled. In theory, this strategy can be used to identify the profiles for any *Drosophila* cell-type in a Drop-seq dataset, providing a corresponding specific GAL4 driver line is available. This is a clear advantage of using *Drosophila* as a model for a brain cell-atlas, because thousands of GAL4 lines are available that direct expression in specific subsets of neurons in the brain

(*Gohl et al., 2011*; *Jenett et al., 2012*; *Tirian and Dickson, 2017*). Cell-specific transgene expression therefore presents the most straightforward means to link single-cell sequencing data to neuro-anatomy and will be very useful to de-orphan our currently 'unannotated' cells.

The extensive collection of cell-specific GAL4 lines were constructed by fusing potential regulatory regions from genes to GAL4 coding sequence (*Jenett et al., 2012*; *Pfeiffer et al., 2008*; *Tirian and Dickson, 2017*). Their expression patterns can therefore indicate elements of the expression of the gene from which they are taken. We also used this property of the GAL4 collections to help us assign single-cell data to specific neurons. For example, we originally suspected that one of the clusters corresponded to ellipsoid body ring neurons because cells in the cluster expressed the *Gad1* marker for GABA-ergic neurons and *Fas2*, an antibody for which is known to label this region of neuropil (*Whited et al., 2007*). To corroborate this assignment to ellipsoid body we asked whether promoter-GAL4 lines constructed from some of the other top new markers for this cluster, such as *Dh31* and *Sox21b*, labeled ellipsoid body ring neurons (*Jenett et al., 2012*; *Tirian and Dickson, 2017*). Indeed, we found that the R20A02 (*Dh31*) and R73A06 (*Sox21b*) GAL4 drivers very specifically express in these neurons, as do other drivers corresponding to the ellipsoid body expressed genes *Dichaete* (R12G08), *SoxN* (R40E11, R41G11, VT004444), *ara* (VT029750) and *gprk2* (R13C06, R13F12). Therefore, by combining the expression of known markers, and querying the specificity of new markers, it is possible to convincingly assign transcriptional profiles to cell-type.

Our initial analyses of the brain cell-atlas also immediately provided a lot of new information that is of functional importance. We focused our first investigations on neurotransmitter usage and the potential for synaptic co-release/co-transmission. These analyses clearly defined the main fast-acting transmitters used by each cell cluster. For example, the KC transmitter was fairly recently determined to be ACh (*Barnstedt et al., 2016*) and consequently all the KC clusters strongly labeled with the cholinergic markers *ChAT* and *VAChT*. The cell-atlas dataset therefore allows one to easily determine the neurotransmitters that a particular cell-type uses, providing the cells of interest can be identified in the cluster plot.

Important questions can also be addressed even without identifying how particular cells appear in the cluster plot. One example is our analyses of potential co-release of multiple fast acting transmitters or fast-acting transmitters with neuropeptides. Our data suggest that a small percentage of neurons might co-release ACh and Glu, or ACh and GABA. Analyzing co-expression of transmitter marker genes and neuropeptide-encoding genes revealed some very interesting and novel findings. We found that some neuropeptides (*Clynen et al., 2010*; *Hewes and Taghert, 2001*; *Nässel and Winther, 2010*), whether expressed in many or only a few cells, are exclusively detected in neurons that use a particular fast-acting transmitter. These correlations suggest a fine relationship between the fast-acting transmitter and neuropeptide-specific modulation. Our co-expression analyses also reveal extensive expression of neuropeptide-encoding and processing genes in monoaminergic neurons. It will be interesting to test whether the apparent heterogeneity of neuropeptide expression in these neurons contributes to their apparent functional specialization (*Aso et al., 2012*; *2014*; *Burke et al., 2012*; *Claridge-Chang et al., 2009*; *Huetteroth et al., 2015*; *Krashes et al., 2009*; *Lin et al., 2014*; *Liu et al., 2012*; *Yamagata et al., 2015*).

The brain cell atlas is of great use to those with a gene-centered view of fly neurobiology. It is now possible to query the atlas and ask how broadly, or cell-specifically, a given gene is expressed. Our initial clustering allows one in some cases to pinpoint the expression to a defined cell-type and region of neuropil. This seems particularly valuable information if one is working with a gene, for example, one that has been implicated in neural disease, but does not know the anatomical context in which it operates. Similarly, if a constitutive mutant fly strain has pleiotropic effects, the expression pattern of the gene can indicate where the different phenotypes might manifest. Moreover, the brain-atlas dataset can provide these answers quickly for multiple genes, and it therefore represents a terrific complement to the usual time-consuming and 'single-gene at a time' approaches, such as technically challenging in-situ hybridization to RNA, generating antibodies, making promoter fusions, or knocking in epitope tags to individual loci. Perhaps most importantly, querying the cell-atlas provides single-cell resolution of gene expression across all the major cell-types in the fly midbrain.

We believe that the potential uses for the cell atlas are almost endless. The data reveal a huge number of new genetic markers for known cell types, and as yet undefined cell types, in the fly brain. Many of these are likely to be functionally important and represent new entry points to guide interventionist experiments to understand how specific molecules operate within the relevant neurons

and networks. Although we focused most of our analyses on neuronal cells, different classes of glia (*Freeman, 2015*) could also be defined.

Our initial analysis was performed on 10,286 of the highest quality cells (≥800 UMIs) from a larger dataset of 19,260 cells. This atlas is effectively a scaffold that can now be continuously updated and expanded as additional cells are collected and sequenced (*Davie et al., 2017*; *Konstantinides et al., 2018*). Our current dataset was derived from cells taken from unique groups of flies, processed on eight separate days, and yet each biological replicate contributed equally to the combined data set. This robustness and reproducibility of the approach is essential to know in order to be able to add data from future experiments to the current cell cluster. Including more cells with a comparably high number of UMIs per cell should increase statistical power and permit further resolution of cell-type. Including more cells with a lower number of UMIs per cell did not improve our analysis.

The current dataset was collected from young flies that were raised under ideal conditions with ample food and water. Future experiments that aim to investigate the impact of changes to the state of the fly, such as age, bacterial infection and starvation, can use the current cell atlas as a foundation to identify changes in expression patterns that may occur in individual cells across the midbrain. Similarly, brains from flies harboring specific mutations can be molecularly characterized using the approach described here, to uncover molecular manifestations of the mutant phenotype.

The fly brain cell atlas described here should also be a valuable resource to researchers working in other animals. Many markers for *Drosophila* cell-type are likely to be conserved in other insects and arthropods, and so will be useful markers for regions of the brain in these animals (*Thoen et al., 2017*; *Wolff and Strausfeld, 2015*). The orthologs of some of these new markers, for example those expressed in subsets of dopaminergic neurons, might also extend to labeling comparable cells in the mammalian brain.

# Materials and methods

**Key resources table**

| Reagent type (species) or resource | Designation | Source or reference | Identifiers | Additional information |
|---|---|---|---|---|
| Genetic reagent (*Drosophila melanogaster*) | MB008B | Bloomington Drosophila Stock Center | RRID:BDSC_68291 | |
| Genetic reagent (*D. melanogaster*) | MB131B | Bloomington Drosophila Stock Center | RRID:BDSC_68265 | |
| Genetic reagent (*D. melanogaster*) | MB461B | Bloomington Drosophila Stock Center | RRID:BDSC_68327 | |
| Genetic reagent (*D. melanogaster*) | uas-mCherry (III) | other | uas-mCherry(III) | lab stock |
| Cell line (*D. melanogaster*) | Drosophila S2 Cells in Schneider's Medium | Gibco, Waltham, MA | R69007 | |
| Cell line (*Spodoptera frugiperda*) | Sf9 cells in Sf-900 III SFM | Gibco | 12659–017 | |
| Sequence-based reagent | Template switch oligo | Sigma, St. Louis, MO | | AAGCAGTGGTATCAACGC AGAGTGAATrGrGrG |
| Chemical compound, drug | Schneider's medium | Gibco | 21720–001 | |
| Chemical compound, drug | FBS | Sigma | F0804 | |
| Chemical compound, drug | penicillin-streptomycin | Gibco | 15070–063 | |
| Chemical compound, drug | Sf-900 III SFM | Gibco | 12658019 | |

*Continued on next page*

*Continued*

| Reagent type (species) or resource | Designation | Source or reference | Identifiers | Additional information |
|---|---|---|---|---|
| Chemical compound, drug | DPBS (calcium and magnesium free) | Gibco | 14190–086 | |
| Chemical compound, drug | Papain | Sigma | P4762 | |
| Chemical compound, drug | Collagenase | Sigma | C2674 | |
| Chemical compound, drug | d(−)−2-amino-5 -phosphonovaleric acid | Sigma | A8054 | |
| Chemical compound, drug | 6,7-dinitroquinoxaline-2,3-dione | Sigma | D0540 | |
| Chemical compound, drug | tetrodotoxin | Abcam, UK | ab120054 | |
| Other | 10 µm CellTrix strainer | Sysmex, Japan | 04-0042-2314 | |
| Other | Fuchs-Rosental hemocytometer | VWR, Radnor, PA | 631–1096 | |
| Commercial assay or kit | Single Cell RNA-Seq system | Dolomite Bio, UK | 3200537 | |
| Chemical compound, drug | Barcoded Beads SeqB | ChemGenes Corp., Wilmington, MA | | |
| Chemical compound, drug | Ficoll PM-400 | VWR | 17-0300-10 | |
| Chemical compound, drug | N-Lauroylsarcosine sodium salt solution | Sigma | L7414 | |
| Chemical compound, drug | QX200 Droplet generation oil for EvaGreen | Biorad, Hercules, CA | 1864006 | |
| Chemical compound, drug | DTT | Life Technologies, Carlsbad, CA | P2325 | |
| Chemical compound, drug | Maxima H Minus Reverse Transcriptase | Thermo Scientific, Waltham, MA | EP0753 | |
| Chemical compound, drug | Exonuclease I | NEB, Ipswich, MA | M0293L | |
| Sequence-based reagent | SMART PCR primer | Sigma | | AAGCAGTGGTATCAA CGCAGAGT |
| Chemical compound, drug | Hifi HotStart Readymix | Kappa Biosystems, Switzerland | KK2602, KK2611 | |
| Chemical compound, drug | Agencourt AMPure XP beads | Beckman-Coulter, Brea, CA | A63880 | |
| Commercial assay, kit | Bioanalyzer High-Sensitivity DNA kit | Agilent, Santa Clara, CA | 5067–4626 | |
| Commercial assay, kit | Nextera XT DNA Sample Preparation Kit | Illumina, San Diego, CA | FC-131–1024 | |
| Sequence-based reagent | New-P5-SMART PCR hybrid | Sigma | | AAT GAT ACG GCG ACC ACC GAG ATC TAC ACG CCT GTC CGC GGA AGC AGT GGT ATC AAC GCA GAG T*A*C |
| Commercial assay, kit | PicoPure™ RNA Isolation Kit | Applied Biosystems, Foster City, CA | KIT0204 | |
| Commercial assay, kit | SuperScript III First-Strand Synthesis SuperMix | Invitrogen, Carlsbad, CA | 18080400 | |
| Commercial assay, kit | QIAquick PCR Purification Kit | Qiagen, Germany | 28106 | |
| Commercial assay, kit | Universal Probe Library system | Roche, Switzerland | 04683633001, 04869877001 | |

*Continued on next page*

Continued

| Reagent type (species) or resource | Designation | Source or reference | Identifiers | Additional information |
|---|---|---|---|---|
| Commercial assay, kit | LightCycler® 480 Probes Master | Roche | 4887301001 | |
| Commercial assay, kit | SMART-Seq v4 Ultra Low Input RNA Kit for Sequencing | Takara Clontech, Japan | 634890 | |
| Commercial assay, kit | TruSeq RNA Library Prep Kit v2 | Illumina | RS-122–2001 | |

## Fly strains

The *Drosophila* strains used were MB008B, MB131B and MB461B (*Aso et al., 2014*), R58E02 (*Pfeiffer et al., 2008*), *w-;+;20XUAS-6XGFP* (*Shearin et al., 2014*) and *w-; +; UAS-mCherry*. Flies were raised at 25°C in 12 hr:12 hr day-night cycles on standard food at 40–50% humidity.

## Cell culture

S2 cells (Gibco, R69007) were grown in Schneider's medium (Gibco 21720–001) supplemented with 10% FBS (Sigma, F0804) and 1% penicillin-streptomycin (Gibco, 15070–063). Sf9 cells (Gibco, 12659–017) were grown in Sf-900 III SFM (Gibco, 12658019). The Master Seed Bank for both Sf9 and S2 cells was tested for contamination of bacteria, yeast, mycoplasma and virus and characterized by isozyme and karyotype analysis by the supplier. All cells were incubated at 25°C. Cells were grown in adherent cultures to confluency. Vessels were gently tapped to detach cells, and supernatants were centrifuged for 10 min at 100 x g. Cells were washed once with 1 x PBS and resuspended in 1 x PBS and subsequently diluted to 200 cells/ul prior to pooling and Drop-seq.

## Brain dissociation and cell collection

The brain dissociation protocol was adapted from previously described methods (*Harzer et al., 2013*; *Nagoshi et al., 2010*). For each day of experiments, 80–100 central brains were individually dissected in ice-cold calcium- and magnesium-free DPBS (Gibco, 14190–086) and immediately transferred into 1 mL toxin-supplemented Schneider's medium (tSM: Gibco, 21720–001 + 50 μM d(−)−2-amino-5-phosphonovaleric acid, 20 μM 6,7-dinitroquinoxaline-2,3-dione and 0.1 μM tetrodotoxin) on ice. Brains were washed once with 1 mL tSM and incubated in tSM containing 1.11 mg/mL papain (Sigma, P4762) and 1.11 mg/mL collagenase I (Sigma, C2674). Brains were washed once more with tSM and subsequently triturated with flame-rounded 200 μL pipette tips. Dissociated brains were resuspended into 1 mL PBS + 0.01% BSA and filtered through a 10 μm CellTrix strainer (Sysmex, 04-0042-2314). Cell concentration was measured using a disposable Fuchs-Rosenthal hemocytometer (VWR, 631–1096) under a Leica DMIL LED Fluo microscope, that also allowed detecting mCherry fluorescence in dissociated KCs. Cells were diluted in PBS + 0.01% BSA up to a concentration of 200 cells/μL. Thus a typical preparation from 80 brains yielded ~2'000'000 single-cells in a volume of 10 mL.

## Drop-seq procedure

Drop-seq was performed as described (*Macosko et al., 2015*), using a Dolomite Bio (Royston, UK) Single Cell RNA-Seq system. Cells were diluted at a concentration of 200 cells/μL into PBS + 0.01% BSA. Barcoded Beads SeqB (ChemGenes Corp., Wilmington, MA, USA) were diluted at a concentration of 200 particles/μL into 200 mM Tris pH 7.5, 6% Ficoll PM-400, 0.2% Sarkosyl, 20 mM EDTA +50 mM DTT.

For each run, 700 μL of cells solution from dissociated brains were loaded into a microcentrifuge tube inside a reservoir connected to a Mitos P-Pump (Dolomite microfluidics, 3200016) set to provide a constant flow of 30 μL/min. The reservoir was placed on a stirring plate and agitation was provided by a stir bar placed inside the reservoir but outside the tube to maintain the cells in suspension while avoiding damaging the cells. 600 μL of beads solution were loaded into a 50 cm sample loop connected to a second Mitos P-Pump set to provide a constant flow of 30 μL/min. The sample loop was used to avoid beads sedimentation while eliminating the need for stirring, thus preventing beads damage. QX200 Droplet Generation Oil for EvaGreen (BioRad, 1864006) was loaded

directly inside a third Mitos P-Pump, set to provide a constant flow of 200 µL/min. Cells, beads and oil flows were connected to a Single Cell RNA Seq Droplet Chip (Dolomite Bio) according to manufacturer's instructions, allowing pairing of single-cells with single-beads and formation of 357 pL droplets of aqueous cell/bead solution in oil. The chip was placed under a Meros High Speed Digital Microscope and Camera with a HLB M Plan Apo 5X objective in order to monitor droplet formation. Droplets were collected in 50 mL Falcon tubes. Reagents were reloaded and Falcon tubes replaced every 15 min.

Droplets were subsequently broken and beads with captured mRNA were washed as described (*Macosko et al., 2015*). In brief, bead-bound mRNA was immediately reverse-transcribed using a Template Switch Oligo (5' – AAG CAG TGG TAT CAA CGC AGA GTG AAT rGrGrG – 3') and Maxima H Minus Reverse Transcriptase (Thermo Scientific, EP0753). cDNA was treated with Exonuclease I (NEB, M0293L) and amplified in multiple 50 µL PCR reactions performed on aliquots of ~2000 beads, using a SMART PCR primer (5' – AAG CAG TGG TAT CAA CGC AGA GT – 3') and Hifi Hot-Start Readymix (Kapa Biosystems, KK2602) for a total of 17 cycles. 10 µL from each PCR reaction were pooled, and amplified cDNA was purified twice, with 0.6X and 1.0X volumes of Agencourt AMPure XP beads (Beckman Coulter, A63880) and quantified on a Bioanalyzer, using a High-Sensitivity DNA kit (Agilent, 5067–4626). From each sample, 2 × 600 pg of amplified cDNA were tagmented using the Nextera XT DNA Sample Preparation Kit (Illumina, FC-131–1024) with New-P5-SMART PCR hybrid (5' – AAT GAT ACG GCG ACC ACC GAG ATC TAC ACG CCT GTC CGC GGA AGC AGT GGT ATC AAC GCA GAG T*A*C – 3') and one of Nextera N701 to N706 oligos. cDNA libraries were purified twice, with 0.6X and 1.0X volumes of Agencourt AMPure XP beads (Beckman Coulter, A63880) and quantified on a Bioanalyzer, using a High-Sensitivity DNA kit (Agilent, 5067–4626). Libraries were pooled together and sequenced on an Illumina HiSeq2500 sequencer using a Custom Read1 primer (5' – GCC TGT CCG CGG AAG CAG TGG TAT CAA CGC AGA GTA C – 3') and standard Illumina Read2 primers. All oligos were synthesized by Sigma, and HPLC purified. Samples from days 1 and 2 were sequenced together, on two separate lanes. Samples from days 3–8 were sequenced together, on three separate lanes.

## Data processing and alignment

Sequencing data was processed as previously described (*Macosko et al., 2015*; *Satija et al., 2015*), following the Drop-seq Computational Protocol v.1.0.1 and using the Drop-seq software tools v.1.13 from the McCarroll lab. Barcodes were extracted and reads were aligned to a combination of the *Drosophila melanogaster* genome release 6.13 (from Flybase.org) and three reference sequences for mCherry and each split-GAL4 transgenes of the flies that were used in this study. For the species mix experiments, reads were also aligned to the *Spodoptera frugiperda* genome (*Kakumani et al., 2014*), available at NCBI GenBank, assembly ASM221328v1. The Flybase v.FB2016_05 September gene names were used for the creation of the Digital Gene Expression (DGE) Matrix.

## t-SNE analysis on whole brain data

Analysis of DGEs was performed with the Seurat 2.1.0 R package (*Macosko et al., 2015*; *Satija et al., 2015*). Cells with less than 200 genes were discarded. Several thresholds for the number of UMIs per cell were tested (see *Figure 1—figure supplement 2*). All results presented here are based on 800 and 10,000 UMIs per cell as lower and higher threshold, respectively. Data was log-normalized and scaled using default options. Variation driven by individual batches was regressed out from the normalized, scaled data. PCA analysis was performed on the data as previously described (*Macosko et al., 2015*). To visualize the data, spectral t-SNE dimensionality reduction was performed, using the first 50 PCAs, as instructed by a Jack Straw resampling test (*Satija et al., 2015*; *Van Der Maaten, 2014*). Clusters were identified by a shared nearest neighbor modularity optimization (*Waltman and van Eck, 2013*), using a resolution of 2.5. Some of these clusters were subsequently manually modified (compare *Figure 1—figure supplement 2B* (unmodified) and *Figure 1B* (modified)). Main markers for each identified cluster were identified as genes with Log2 FC $\geq$1 and a p-value of $p<0.01$ (after Bonferroni correction).

## Co-expression analysis

Gene co-expression was assessed by calculating the Pearson product-moment correlation of the log-normalized, scaled expression values using R. For the radar plots, the number of cells expressing each neuropeptide of interest and in addition either *VACht* (to identify cholinergic cells), *VGlut* (glutamatergic), *Gad1* (GABA-ergic) or combinations of the three were calculated and normalized to the total number of cells expressing each neurotransmitter.

## t-SNE analysis on PNs and monoaminergic neurons (re-clustering)

DGE columns corresponding to cells belonging to either PN or monoaminergic clusters were used for PCA analyses. For re-clustering of monoaminergic neurons, a selection of known markers (*ple*, *DAT*, *SerT*, *Trh*, *Vmat*, *Oamb*, *Ddc*, *Tdc2* and *Tbh*) was used as input for PCA analysis. In both cases, the first 6 PCAs were used for re-clustering, which was performed as above.

## Quantitative PCR

Midbrains were dissected from flies expressing *mCherry* under the control of the MB008B, MB131B or MB461B GAL4 drivers, dissociated and filtered as above. DAPI was added to the cell suspension as a marker for dead cells or cells with compromised membrane (*Kubista et al., 1987*), and only DAPI- cells were selected. Filtered cells were sorted with a MoFlo Astrios (Beckman Coulter) and mCherry +cells were collected from each genotype, in biological triplicates. Total RNA was extracted from these cells using the PicoPure RNA Isolation Kit (Applied Biosystems, KIT0204) according to manufacturer's instructions. mRNA was then retrotranscribed using the SuperScript III First-Strand Synthesis SuperMix (Invitrogen, 18080400) according to manufacturer's instructions. Obtained cDNA was pre-amplified with the KAPA HiFi HotStart ReadyMix (Kapa Biosystems, KK2611), using 0.4 µM of each primer (see primers list in *Figure 2—source data 1*). Pre-amplification protocol was as follows: 98°C, 2'; 18X [98°C, 20'; 60°C, 30', 72°C, 30']; 72°C, 2'. Pre-amplified cDNA was purified with the QIAquick PCR Purification Kit (Qiagen, 28106). qPCR was performed in a Light-Cycler 480 Instrument II (Roche, 05015243001) using the Universal Probe Library system (UPL; Roche, 04683633001 and 04869877001). Each 10 µL reaction contained 2.4 µL of pre-amplified cDNA, 0.4 µM of each primer (designed with Roche Assay Design Center), 0.2 µM of the corresponding UPL probe (*Figure 2—source data 1*), and 5 µL LightCycler 480 Probes Master (Roche, 4887301001). Cycles were as follows: 95°C, 10'; 45X [95°C, 10'; 60°C, 30'; Fluorescence acquisition; 72°C, 1']. Quantification was performed with the comparative 2-ΔΔCt method (*Livak and Schmittgen, 2001*), using *SdhA* as a housekeeping gene (*Ling and Salvaterra, 2011*; *Treiber and Waddell, 2017*). For each biological replicate, expression levels of each gene were normalized to the KC population with the highest expression level, and subsequently averaged. Genes for which qPCR signal was not consistently observed across all samples and replicates were not included in the analysis.

## RNA-sequencing of PAM-DA neurons

Central brains from flies expressing a brighter, hexameric GFP (20xUAS-6xGFP; [*Shearin et al., 2014*]) specifically in PAM-DA neurons under the control of the R58E02 GAL4 line (*Pfeiffer et al., 2008*) were dissected and dissociated as above. FACS was performed as above, and both GFP +and GFP- cells were collected. Cells were lysed, and their mRNA was retro-transcribed and amplified (17 cycles) using the SMART-Seq v4 Ultra Low Input RNA Kit for Sequencing (Takara Clontech, 634890), according to manufacturer's instructions. Biological triplicates were made for each sample. cDNA libraries were generated with TruSeq RNA Library Prep Kit v2 (Illumina, RS-122–2001), sequenced on an Illumina HiSeq4000 sequencer. Results were analyzed using the Tuxedo RNA-seq pipeline (*Trapnell et al., 2012*).

## Acknowledgements

We thank the Bloomington Stock Center for flies. We are grateful to Michal Maj for help with FACS, Paola Cognigni for help with R and statistics, and other members of the Waddell lab for discussion. RNA-sequencing was performed at *MacroGen* Inc. and the Oxford Genomic Centre. CT was supported by a Wellcome Trust PhD studentship and SW is funded by a Wellcome Trust Principal

Research Fellowship in the Basic Biomedical Sciences (200846/Z/16/Z) and the Bettencourt-Schueller Foundation.

## Additional information

### Funding

| Funder | Grant reference number | Author |
|---|---|---|
| Wellcome | 200846/Z/16/Z | Scott Waddell |
| Fondation Bettencourt Schueller | | Scott Waddell |
| Wellcome | | Vincent Croset<br>Christoph D Treiber |

The funders had no role in study design, data collection and interpretation, or the decision to submit the work for publication.

### Author contributions

Vincent Croset, Conceptualization, Formal analysis, Validation, Investigation, Visualization, Methodology, Writing—original draft, Writing—review and editing; Christoph D Treiber, Conceptualization, Data curation, Formal analysis, Validation, Investigation, Visualization, Methodology, Writing—original draft, Writing—review and editing; Scott Waddell, Conceptualization, Supervision, Funding acquisition, Writing—original draft, Writing—review and editing

### Author ORCIDs

Christoph D Treiber (iD) http://orcid.org/0000-0002-6994-091X
Scott Waddell (iD) https://orcid.org/0000-0003-4503-6229

### Decision letter and Author response

Decision letter https://doi.org/10.7554/eLife.34550.025
Author response https://doi.org/10.7554/eLife.34550.026

## Additional files

### Supplementary files

• Transparent reporting form
DOI: https://doi.org/10.7554/eLife.34550.021

### Data availability

Sequencing data have been deposited in SRA under accession code SRP128516. Digital Expression Matrix is provided as supplementary file.

The following dataset was generated:

| Author(s) | Year | Dataset title | Dataset URL | Database, license, and accessibility information |
|---|---|---|---|---|
| Croset V, Treiber CD, Waddell S | 2018 | Data from: Cellular diversity in the Drosophila midbrain revealed by single-cell transcriptomics | https://www.ncbi.nlm.nih.gov/sra/SRP128516 | Publicly available at the NCBI Sequence Read Archive (accession no: SRP128516) |

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
