## [Decision Letter]

Thank you for submitting your article "Cellular diversity in the *Drosophila* midbrain revealed by single-cell transcriptomics" for consideration by *eLife*. Your article has been reviewed by three peer reviewers, one of whom, Mani Ramaswami, is a member of our Board of Reviewing Editors, and the evaluation has been overseen by K VijayRaghavan as the Senior Editor.

The reviewers have discussed the reviews with one another and the Reviewing Editor has drafted this decision to help you prepare a revised submission.

Summary:

The authors describe the use of a very exciting new technology, Drop-seq (Macosko et al., 2015) to determine single-cell transcriptomes from thousands of *Drosophila* midbrain cells. Based on differences in their transcriptional profiles, the authors group individual cells into dozens of clusters, and thereby characterize the diversity of cell-types in the brain. Within each cluster they map potential correlations in expression of neuromodulators, neuromodulator receptors, and fast neurotransmitters. And most usefully, they use expression of known marker genes to assign several of the clusters to specific anatomically, and in some cases functionally, analysed neuronal cell types. Thus, they establish a first draft database of gene expression in cell-types of the *Drosophila* brain.

Overall, this is an extremely interesting study, describing the application of DropSeq to learn new biology in an intensively studied and important biological system Using newly gained resolution on cell-transcriptome heterogeneity, the work teases apart the cellular complexity of the *Drosophila* simple brain. Although the analysis largely reports a list of correlations and co-expression percentages, the data provide a level of resolution and a global description of highly interesting and well-studied cells that was not hitherto possible. Using their original data and internal controls, the authors recapitulate several known biological insights gained over years of detailed genetic analyses and deliver several new insights that will be of broad interest to groups studying an advanced system of perception, learning and memory as well as neurobiologists in general.

However, in many places, the work could go further. The paper could be greatly enhanced to provide deeper discussion of caveats in the interpretation of the data, as well as a few additional lines of control/ confirmative information necessary to clarify levels of confidence in the details of the interesting observations. The following issues should be addressed:

Essential revisions:

1) Methods are cryptic in part and need to be elaborated both for clarity and to evaluate intrinsic limitations of the analysis. How did the authors define the specific cell types/brain regions presented in Figure 1B (e.g. fat body)? How were KCs in Figure 7C identified? Which cells are considered for the analyses of nAChRs (Figure 8)? Which GAL4 is used for the optogenetic activation of neurons in Figure 8?

2) In this context, despite the nice control analysis showing that S2 cell and army-worm cell transcriptomes are well separated, there is still a problem that unlike cultured cells, following disruption of a brain even with "trituration," cells may not be easily isolated intact. It would be useful to provide simple images of the cell suspensions to show whether one sees nuclei, single intact cells and/or occasional clumps. The singlet/doublet estimate based on cultured cells is not necessarily a good, transferable measure for dissociated brains where cell sizes, cell adhesion and cell lysis issues can be different and this needs to be addressed in the text and with some data (that may be on hand). Regarding cell lysis: did the authors consider mild fixation and see how this affected sample morphology?

3) The paper would be greatly improved with some more direct functional evidence beyond clustering and identification of highly variable genes as markers. Could the authors provide more evidence of some of the newly found markers via in situ hybridization, antibody staining or reporter lines, for example? The authors should comment explicitly on some apparent discrepancies: genes like fruitless and sNPF have been previously reported to be expressed outside the mushroom body. Data from this study appear to show them to be mushroom body (a/b neuron)-specific. Furthermore, the dopamine transporter (DAT) is expected to be highly expressed in dopamine neurons, DropSeq data indicates that the expression is specifically in a'/b' neurons. What are the implications of these apparent discrepancies to interpretation of the data?

4) The use of literature-based marker genes to specify cell types is certainly a powerful way, but the assignment is not always convincing. Cluster identification to PNs is for example questionable, as many of these marker genes are expressed outside PNs (e.g. Certel et al., 2000 for acj6). Unexpected gene expression patterns (e.g. Ddc expression in PNs and KCs (subsection “Dopamine metabolism”) may also be due to misannotation. This should be addressed experimentally and/or acknowledged in the text.

Another example is insulin-like peptide 6 expressing in glia based on the co-expression with the transmitter synthesizing enzymes, but no description about glia-specifying genes like repo. Since cell type annotation is critical, various statements should be moderated to acknowledge the need for further confirmatory experiments. In general, the authors should also consider how reasonable it is to call expression in a binary fashion, instead of in a graded fashion, as cut-offs can be tricky when data are shallow. The text in this regard may needs to be clearer and the presentation of data may need to be adjusted.

---

## [Author Response]

Essential revisions:

1) Methods are cryptic in part and need to be elaborated both for clarity and to evaluate intrinsic limitations of the analysis. How did the authors define the specific cell types/brain regions presented in Figure 1B (e.g. fat body)?

As mentioned in the text we used the list of markers presented in Figure 1—source data 2 to assign the identity of each cluster. This list contains all genes that we find to be significantly over-expressed in each cluster, compared to the rest of the brain. For clarity we have modified the text to mention that “We then used the published expression patterns for many of these genes to assign identity to several clusters”. We have also added relevant references to Figure 1—source data 2.

How were KCs in Figure 7C identified?

As described for Figure 2, we used expression of the transgenically expressed marker mCherry (in αβ neurons) and a number of known KC subtype markers to identify these cells. For example, *eyeless* and *Dop1R2* expression clearly identify the three clusters corresponding to KCs. Many other known KC markers are expressed in these neurons (see Figure 1—source data 2, including sNPF in αβ and γ but not α′β’ and *trio* in α′β’ and γ but not αβ – Figure 2E). We used all the cells contained in these three KC clusters for the analysis shown in Figure 7C (now 8C). We have now added this detail to the previously missing legend for this figure.

Which cells are considered for the analyses of nAChRs (Figure 8)?

We used all 10,286 cells described in this paper for this analysis. The text was modified as follows: “We therefore analyzed the co-expression of nAChR subunits using all cells from our Drop-seq dataset”.

Which GAL4 is used for the optogenetic activation of neurons in Figure 8?

We assume the reviewers are referring to the top panel of Figure 9 in our original manuscript (now Figure 10)? If so, as mentioned in the text, these data were taken from the Chen et al., 2016 paper from the Rosbash lab. In this paper, the authors used the pan-neuronal elav-Gal4 driver to drive ChR2-XXL expression. We have now modified the text to mention that the optogenetic activation in that study was pan-neuronal.

2) In this context, despite the nice control analysis showing that S2 cell and army-worm cell transcriptomes are well separated, there is still a problem that unlike cultured cells, following disruption of a brain even with "trituration," cells may not be easily isolated intact. It would be useful to provide simple images of the cell suspensions to show whether one sees nuclei, single intact cells and/or occasional clumps. The singlet/doublet estimate based on cultured cells is not necessarily a good, transferable measure for dissociated brains where cell sizes, cell adhesion and cell lysis issues can be different and this needs to be addressed in the text and with some data (that may be on hand). Regarding cell lysis: did the authors consider mild fixation and see how this affected sample morphology?

Making sure that the solution used for Drop-seq mostly contains single cells is indeed very important. As mentioned in the Materials and methods section, we always filter the cell suspension through a 10 µm (about the size of the cell body of a typical *Drosophila* neuron) cell strainer prior to Drop-seq and FACS experiments. We are therefore confident that no clumps or particles bigger than 10 µm are included in the cell suspension. We also always verify the quality of our preparation while counting the cells after filtration. As described in the Materials and methods section, we have also discarded all barcodes containing >10,000 UMIs, as these are likely to have arisen from doublets. Although fragments from lysed cells and other smaller particles may be able to pass through the strainer, these are unlikely to contain significant amounts of mRNA and would thus fall below the 800-UMIs threshold of cells that are included in our analysis. We have now added some text to mention the filtration process: “Larger tissue fragments were removed by filtration, and the eluant was inspected under a microscope to confirm the presence of single cells and the absence of clumps”.

Despite these precautions, we cannot exclude that, in some occasions, two neighboring cells could remain attached and make their way through the filter and into the analysis. The extent of this phenomenon is extremely difficult to measure, as it depends on the nature of the bonds between each pair of cells. For example, our data suggests that fragments of glia could, in some very rare cases, remain attached to neurons (31 out of 9955 (0.3%) cells expressing neuronal markers also express glial markers – see new paragraph about Cluster M, paragraph three subsection “Identification of glia and astrocytes”). Another possible example that demonstrates that this phenomenon may be possible, but very rare, is that we only find two α′β’ and five γ KCs expressing the αβ-driven *mCherry* (Figure 2B), and only two α′β’ KCs that express the αβ- and γ-specific *sNPF* (Figure 2E), even although the somata of these different KC cell types are all tightly interspersed in the brain, and therefore could remain together after dissociation.

As for cell survival, we observed in the FACS experiments included in this manuscript that the fraction of DAPI+ cells (DAPI can only penetrate cells with compromised membrane) was always very low. Since the dissociation protocols for FACS and Drop-seq experiments were exactly the same, we are confident that a similarly low number of dead cells was present in our Drop-seq experiments. For completeness of argument, we actually only identified 24 cells of the 10,286 in our dataset that express the two apoptosis markers *hid* and *reaper*.

Since we estimate that cell survival was high, we did not consider including a fixation step (e.g. methanol method, Alles et al., 2017). However we appreciate that this might be useful to use with more fragile cells (perhaps if looking at pupal brain?), for cases where cells need to be kept for a few days, or transported, prior to Drop-seq, or for comparison of gene expression between samples when timing is particularly important.

Lastly, we could add an image of the cell suspension if the reviewers insist that it’s necessary. However, given the enormous number of cells in the analysis, we believe it would not be convincing and would perhaps be seen as simply cherry-picking an appropriate view.

3) The paper would be greatly improved with some more direct functional evidence beyond clustering and identification of highly variable genes as markers. Could the authors provide more evidence of some of the newly found markers via in situ hybridization, antibody staining or reporter lines, for example?

We now include new data that confirms the expression patterns of several of the genes identified in Figure 2F. We used FACS to isolate neurons from each KC subset, and performed qPCR experiments on mRNA extracted from these cells (Figure 2—figure supplement 1). This recapitulates the gene expression observed in the Drop-seq data with outstanding accuracy, and we believe it nicely demonstrates that Drop-seq can be used to reliably identify novel cellular markers. We now mention this experiment in the main text and Materials and methods section, and have added a figure legend for this new supplementary figure.

We previously mentioned a few reporter lines in the discussion of the manuscript, whose expression supports our data assignment to the ring neurons of the ellipsoid body. From the FlyLight collection the R20A02 (*Dh31*) and R73A06 (*Sox21b*) GAL4 drivers very specifically express in the EB ring neurons neurons. We have now added mention of a few more Gal4 lines, that label the ring neurons.

The authors should comment explicitly on some apparent discrepancies: genes like fruitless and sNPF have been previously reported to be expressed outside the mushroom body. Data from this study appear to show them to be mushroom body (a/b neuron)-specific. Furthermore, the dopamine transporter (DAT) is expected to be highly expressed in dopamine neurons, DropSeq data indicates that the expression is specifically in a'/b' neurons. What are the implications of these apparent discrepancies to interpretation of the data?

We think we have unfortunately misled the reviewers with the display of the data in this figure. There is no discrepancy with published work with respect to these genes. The rightmost (grey) column in Figure 2F represents global expression across all non-MB neurons. The purpose of this column is simply to indicate how broadly these genes are expressed outside the MB. Genes such as *fru, sNPF* and *DAT* are indeed found outside of the MB (see for example Figure 3 for *sNPF* and Figures 7-8 for *DAT*). However these genes are expressed in a limited number of the larger population of ‘other’ neurons, hence the “slim” shape of the Violin plots for these genes. This contrasts with other genes like *bol* or *Rbp6*, which are broadly expressed across the whole brain, and that we find to be specifically absent from α'β' and αβ, respectively. We have now clarified the legend of Figure 2F to deal with this confusion.

4) The use of literature-based marker genes to specify cell types is certainly a powerful way, but the assignment is not always convincing. Cluster identification to PNs is for example questionable, as many of these marker genes are expressed outside PNs (e.g. Certel et al., 2000 for acj6). Unexpected gene expression patterns (e.g. Ddc expression in PNs and KCs (subsection “Dopamine metabolism”) may also be due to misannotation. This should be addressed experimentally and/or acknowledged in the text.

We are confident that the neurons that we consider as olfactory PNs in our report at least encompass most of these cells, because these clusters were the only two that strongly co-expressed the well-described PN markers *acj6* and *cut*. We have added a sentence in order to make this clear, and to acknowledge the possibility that these clusters could indeed contain other neurons, that express these two genes but are actually not olfactory PNs, mentioning the Certel et al. paper. Our iterative clustering however also shows expression of additional markers, known to be expressed in subsets of PNs.

In addition to *eyeless, Dop1R2, trio, sNPF* and *Fas2*, that are mentioned in Figure 2, we found a host of other known KC markers expressed specifically in the three clusters that we assign as KCs. These include *rutabaga, Oamb, Pka-R2, Pka-C1, dachsund*, etc, most of which have been shown to play important roles in the development and/or function of the Mushroom Body, and to be robustly expressed in these cells. We are therefore very confident that these cells are KCs. The exhaustive list of these genes, and relevant references linking their expression in the MB, are all mentioned in Figure 1—source data 2. We appreciate that these analyses can identify some things that are unexpected and that perhaps others might find unsettling, based on prior expectations. However, since *Ddc* is robustly expressed in the same cell clusters as many markers for KCs and PNs, we believe the most parsimonious explanation is that *Ddc* is indeed expressed in these cells, as shown in Figure 8B.

Another example is insulin-like peptide 6 expressing in glia based on the co-expression with the transmitter synthesizing enzymes, but no description about glia-specifying genes like repo. Since cell type annotation is critical, various statements should be moderated to acknowledge the need for further confirmatory experiments.

We appreciate that we fell short with this section and have now included a more detailed analysis and new figure (Figure 4) detailing the assignment of glia. We have also added a new supplementary figure (Figure 4—figure supplement 1) that highlights the cells in the tSNE plot that express the neuronal markers *elav* and *nsyb* as well as *repo, nrv2* (glia markers) and *alrm* (astrocyte marker). These non-neuronal clusters express large numbers of other glial-specific genes and show minimal labelling for genes involved in defining fast-acting neurotransmitter labelling in Figure 5A. This new supplementary figure also demonstrates that *Ilp6* is indeed co-expressed with these glial genes and not with neuronal markers, thus confirming its expression in glia. We have added the following sentence in the appropriate section of the manuscript: “…whilst Ilp6 expression is strongly correlated with cells that do not express neurotransmitter markers, but that are positive for the glia-specific genes *repo* and *nrv2* (Figure 4—figure supplement 1).

In general, the authors should also consider how reasonable it is to call expression in a binary fashion, instead of in a graded fashion, as cut-offs can be tricky when data are shallow. The text in this regard may needs to be clearer and the presentation of data may need to be adjusted.

Unfortunately there has also been a misunderstanding here. We totally agree with the reviewers that defining cut-offs for expression can be tricky. For this reason, we actually did not binarize our data in the majority of our analyses. The co-expression analyzes presented in Figures 9 and 10 use non-binary expression values, and of the 18 plots that visualize expression of specific genes in our manuscript (24 in the revised version), only 2 contain data that was binarized (Figure 1—figure supplement 4B and Figure 5A). In Figure 1—figure supplement 4B it seemed appropriate to binarize expression of *roX1* because we observed that *roX1* expression exhibited a biphasic distribution of values. In addition, since *roX1* was previously reported to be male-specific and therefore absent in females, we decided that it would be helpful, in this particular case, to show *roX1* expression in a binary manner. In Figure 5A, we decided to threshold the expression for graphical reasons. To reiterate, for all but two of the analyses, the data have not been called in a binary manner.